# A Survey of Autonomous Vehicle Behaviors: Trajectory Planning Algorithms, Sensed Collision Risks, and User Expectations

**DOI:** 10.3390/s24154808

**Published:** 2024-07-24

**Authors:** Taokai Xia, Hui Chen

**Affiliations:** School of Automotive Studies, Tongji University, Shanghai 201804, China; xiataokai@tongji.edu.cn

**Keywords:** autonomous vehicle, motion planning, collision risk, personalization

## Abstract

Autonomous vehicles are rapidly advancing and have the potential to revolutionize transportation in the future. This paper primarily focuses on vehicle motion trajectory planning algorithms, examining the methods for estimating collision risks based on sensed environmental information and approaches for achieving user-aligned trajectory planning results. It investigates the different categories of planning algorithms within the scope of local trajectory planning applications for autonomous driving, discussing and differentiating their properties in detail through a review of the recent studies. The risk estimation methods are classified and introduced based on their descriptions of the sensed collision risks in traffic environments and their integration with trajectory planning algorithms. Additionally, various user experience-oriented methods, which utilize human data to enhance the trajectory planning performance and generate human-like trajectories, are explored. The paper provides comparative analyses of these algorithms and methods from different perspectives, revealing the interconnections between these topics. The current challenges and future prospects of the trajectory planning tasks in autonomous vehicles are also discussed.

## 1. Introduction

Over the past decade, there has been a rapid development of autonomous vehicles (AV), garnering attention from various sectors including academia, industry, and governments. As the autonomous driving systems (ADSs) in these vehicles continue to advance, they hold the potential to revolutionize transportation, yet they are also met with both anticipation and skepticism at the same time.

Autonomous vehicles have demonstrated the capability to avert potential collisions in traffic environments with human errors. They were reported to reduce accidents and enhance road safety in previous investigations [1]. Even with the lane-keeping assistance (LKA) system, a low-level automated driving assistance system (ADAS), which falls short of ADSs, there remains a substantial reduction in the likelihood of potential fatal head-on crashes [2]. Autonomous vehicles are also promising to alleviate traffic congestion and improve transportation efficiency, especially in populated urban areas with dense traffic flows [3,4].

However, despite these advantages, the widespread application of autonomous vehicles still faces significant challenges and issues regarding safety and user experience. On the one hand, the existing surveys indicate that the the current autonomous vehicles continue to be involved in a notable number of traffic safety incidents in the real world [5]. On the other hand, human drivers usually perceive traditional manual driving as safer and more reliable than the currently available autonomous driving experiences [6]. Research has also observed that drivers’ mental cognitive load will increase when traveling by a commercial autonomous vehicle [7]. The development and update of trajectory planning algorithms, which determine the behaviors and motion of autonomous vehicles directly, should take all these issues into account.

At present, there are many studies in the literature that review and analyze various autonomous driving trajectory planning algorithms. These reviews typically focus on the categorization of trajectory planning algorithms, as well as practical research application cases, aiming to differentiate the characteristics among different algorithms.

The classifications employed in these literature reviews typically include the sampling-based algorithms, the optimization-based algorithms, the interpolating curves, or primitives [8,9,10,11,12,13]. The sampling-based algorithms can be further categorized into graph-search-based algorithms and random-sampling algorithms. The optimization-based algorithms are typically categorized into shooting algorithms and collocation algorithms [14]. Parts of the reviews listed out trajectory planning algorithms based on artificial potential fields or virtual force fields, which are usually optimization-based algorithms too [9,10,13]. We discuss these field-based risk estimation methods in a section of this paper separated from the trajectory planning algorithms themselves more systematically. Model predictive control algorithms were also specifically focused on due to their flexibility, which is also present with shooting optimization-based algorithms [9,15,16]. Bio-inspired swarm intelligence algorithms and other types of rule-based algorithms were also introduced [10,12,13]. The algorithms mentioned above convert trajectory planning tasks to mathematical problems with well-defined forms and clear explanatory representations. They are used in combination with other autonomous driving algorithm modules and have clear hierarchical relationships with them, hence also being referred to as pipeline frameworks or modular approaches.

With the evolution of deep learning technology, several review articles have synthesized end-to-end trajectory planning methodologies, such as imitation learning grounded in supervised learning, reinforcement learning leveraging unsupervised learning, and parallel learning integrating confusion learning [17,18]. Other reviews have summarized emerging multimodal large language models for planning [19]. These methodologies form unified machine learning tasks aimed at generating control commands directly from raw input information. While they exhibit significant potential, their black-box nature poses challenges in determining the underlying causes of unexpected behaviors and implementing relevant fixes for commercial deployment, which deserve more systematic research in the future. Therefore, this paper primarily concentrates on local trajectory planning algorithms based on traditional modular approaches.

The autonomous driving system is a complex system, where the selection of planning algorithm types is just one of the influencing factors affecting the final trajectory planning results in complex traffic environments. The algorithm realization details and their relationship with other vehicle modules are equally crucial. Regarding driving safety, there are reviews about the limitations of the current autonomous vehicles and their impacts on transportation accidents and collision risks [5]. Regarding the user acceptance problem, there are reviews about the social interaction behaviors of autonomous vehicles [4] and how to realize personalized automated driving behaviors based on human trajectory data [17,20,21].

However, the aforementioned reviews exhibit several limitations: they often discuss details in different research cases or types of planning algorithms without thoroughly comparing their characteristics; they address the potential risks and issues faced by the current autonomous driving systems without detailing how autonomous driving algorithms specifically mitigate these risks; and they emphasize the user experience and personalized behavior in autonomous driving systems yet provide limited discussion on the relationship between driver behavior and planning algorithms. To address these gaps, this paper re-examines the issues of the collision risk estimations and user expectations of autonomous driving system behavior from the perspective of trajectory planning algorithm design and explores the interconnections between these algorithms and methods.

The contributions are outlined below, as shown in Figure 1:It presents a taxonomy of the trajectory planning algorithms for autonomous vehicles, elucidating the distinctive characteristics of various algorithmic types.It discusses collision risk estimation methods that can parallelly coexist with or are directly integrated into the trajectory planning algorithms across different planning scenarios.It examines diverse approaches to incorporate the data derived from human driver behaviors into the trajectory planning algorithm development process, exploring their implications on user expectations.

In Section 1, we introduce the advantages and challenges associated with autonomous vehicles, along with a concise overview of the literature reviews relevant to trajectory planning. Section 2 delves into various trajectory planning algorithms, offering a comparative analysis. In Section 3, we explore and compare the different risk estimation methods for trajectory planning tasks. Section 4 scrutinizes the methods for utilizing human data in trajectory planning, alongside the discussions of their impact on user expectations. Section 5 discusses the challenges and future perspectives of trajectory planning for autonomous vehicles. In Section 6, the conclusions are provided.

## 2. Trajectory Planning Algorithms in Autonomous Vehicles

In this section, works of graph-search-based, random-sampling-based, interpolating curves, shooting optimization, collocation optimization, and end-to-end data-driven algorithms for the trajectory planning tasks of autonomous vehicles are reviewed. It is worth noting that the graph-search-based and random-sampling-based algorithms discussed below are also widely applied to global path planning tasks in autonomous driving navigation tasks. However, this article primarily focuses on their applications in collision-free local trajectory planning tasks. These tasks can be accomplished using similar algorithms but with different problem definitions. Additionally, these types of algorithms are also used in the trajectory planning of other controlled objects such as unmanned ground vehicles, unmanned surface vehicles, and unmanned aerial vehicles. Due to space limitations, the properties of these algorithms are analyzed and compared only within the scope of the autonomous vehicle applications in this article. Furthermore, different types of trajectory planning algorithms are often combined in practical applications to address complex driving scenarios.

### 2.1. Graph-Search-Based Trajectory Planning

Graph search algorithms are based on graph theory, solving trajectory planning problems by searching in graphs. It is necessary to define the method of discretizing the motion state space before graph searches, typically applicable to trajectory planning problems in road traffic environments with relatively regular and deterministic states. This graph can consist of discrete nodes or grids in spatial or temporal dimensions, and edges representing specific executable actions. The core idea is to search for the path with the minimum costs in the graph. These costs are usually the travel distance along the path or trajectory but may also include factors such as smoothness and energy consumption in some studies.

One of the most widely used graph search algorithms is the Dijkstra algorithm [22,23,24], which aims to find the desired path from the starting node to the goal nodes by progressively expanding the edges, maintaining a set of nodes awaiting exploration during searches. At each step, it selects the node with the smallest cost from the starting node for expansion. The primary limitations of this algorithm lie in its inability to handle graphs with edges of negative weights and its tendency to equally explore edges in different directions, often disregarding the relationship between new nodes or edges and the goal nodes. These limitations may lead to computationally intensive problems. However, the algorithm’s flexibility in designing costs allows it to effectively address multi-objective trajectory planning problems without the requirement for the differentiability or continuity of optimization objectives. The uniform-cost search (UCS) algorithm shares nearly identical principles with the Dijkstra algorithm but can provide feasible suboptimal solutions before reaching the final optimal solution [25], making it suitable for planning tasks with limited computational resources. Figure 2 demonstrates the spatio-temporal search graph utilized in [25] for trajectory planning tasks on structured roads. The problems involved in spatio-temporal trajectory planning tasks are further discussed in Section 2.7.

To address the computational efficiency issues of Dijkstra algorithm, popular A* (A-star) algorithms are formed by adding appropriate heuristic terms to the search cost, significantly reducing the number of edge and node accesses during the search process [26,27,28,29]. The added heuristic terms provide estimated cost information between the current search node and the endpoint, thereby accelerating the search process. When the heuristic term consistently represents a lower bound on the actual cost, it ensures the optimality of the search results. However, when the heuristic term may exceed the actual cost, only more greedy suboptimal solutions can be obtained. It is worth noting that, for relatively simple optimization objectives and costs, such as shortest distance, it is convenient to set heuristic terms and utilize A* and its derivative algorithms. However, for graph search planning problems with more complex optimization objectives, designing heuristic terms is often challenging, necessitating the use of exhaustive search algorithms without heuristics and Dijkstra-like algorithm [25,30,31].

There are also several derivative improved versions of the A* algorithm, such as the hybrid A* (HA*) algorithm [32,33], which offers the following enhancements. It adopts a discretization method of planning space using spatial grid points, expands neighbor nodes based on vehicle kinematic constraints, and utilizes a combination of two types of heuristics accordingly, allowing for full consideration of vehicle kinematic constraints and trajectory feasibility. However, a drawback of this algorithm is that it can only yield suboptimal solutions within the feasible discrete space. The more recently proposed traversability hybrid A* (THA*) algorithm further uses estimated traversability to optimize the quality of the planned path and outperformed original hybrid A* algorithm in long distance planning tasks [34].

Adjusting the weights of heuristic terms can also improve the efficiency of A* algorithms. Weighted A* algorithm [35] adjusts the weights of heuristic terms, sacrificing optimality of solutions to enhance search efficiency.

Other A* variants also enhance search efficiency by reusing existing historical search information. The anytime tree-restoring weighted A* (ATRWA*) algorithm [36] reduces search time by reusing search information while sacrificing some optimality. The D* algorithm [37] is capable of planning in partially known and dynamically changing environments, avoiding rebuilding search maps at each adjacent planning step, and instead performs rapid replanning based on existing search results and information about environmental changes. The D* Lite algorithm [38] retains the excellent characteristics of the D* algorithm while having a simpler mathematical implementation. The lifelong planning A* (LPA*) algorithm [39], proposed by the same authors, also possesses similar properties. A new and improved anytime dynamic A* (iADA*) algorithm with path optimization during vehicle movement is claimed to be faster than D* Lite and other similar algorithms [40].

The graph-search-based algorithms discussed and analyzed above are summarized in Table 1.

### 2.2. Random-Sampling-Based Trajectory Planning

Unlike graph-search-based trajectory planning algorithms, random-sampling-based trajectory planning algorithms do not require predefining the discretization of the planning state space. Instead, they incrementally explore the state space and perform planning tasks by randomly sampling within the state space. Such search algorithms are better equipped to handle irregular geometric properties and partially unknown states in the planning space. They also circumvent the difficulty of setting discrete precision for specific planning tasks encountered by graph-search-based search algorithms.

Random sampling algorithms are still search-based planning algorithms, requiring the construction of a search graph or a search tree during the planning process. Algorithms based on search graphs include the Probabilistic Roadmap (PRM) algorithm [41] and its derivative, the PRM* algorithm [42], searching paths between the starting and end points by the randomly built maps in the state space. The primary challenge faced by PRM algorithms is the complexity of obstacle collision detection and the significant computational time it consumes. In practical applications, a “lazy” approach is often adopted, deferring collision detection steps by first searching for trajectories and then verifying their feasibility [43]. In contrast, Rapid-exploring Random Tree (RRT) algorithms [44], and other search-tree-based random-sampling algorithms derived from them, have gained more widespread application. PRM and RRT algorithms are probabilistically complete. That is, under the condition of a sufficient number of samples, planning problems with feasible solutions can be solved by these algorithms. Commonly used improvement algorithms include the RRT-connect algorithm [45] or Bi-RRT algorithm [46], which simultaneously generate two search trees from the starting and end nodes, as demonstrated in Figure 3, significantly improving the search efficiency.

Some improved versions of RRT algorithms further focus on achieving asymptotic optimality. This means that, with a sufficient number of samples, the obtained feasible solutions can be continuously improved to approach the optimal solution. Compared to the original RRT, the RRT* algorithm [42,47] changes the way newly added nodes search for parent nodes in the search tree and provides rewire steps for existing nodes, ensuring asymptotic optimality. The informed RRT* algorithm [48] accelerates the process of improving the optimal solution by limiting the sampling range of the state space based on the path length information of the current most feasible solution.

The kinodynamic RRT* [49] and kinematic constrained bidirectional RRT* [50] algorithms improve the generation of new nodes by introducing forward-reachable sets or kinematic constraints between nodes, allowing for consideration of the physic constraints of the controlled object during the search process.

However, the vast majority of nodes sampled in RRT and its derivative algorithms, as well as the edges constructed in the search tree, are redundant and do not contribute to the final result, leading to a waste of computational resources. Some random sampling algorithms based on Edge-implicit Random Geometric Graphs (RGG) [51] address this issue. The FMT* algorithm [52] applies the Dijkstra algorithm to RGG, often achieving faster convergence rates than the RRT* algorithm. Combining the heuristic terms from search graphs and the search tree generated by random-sampling give rise to the BIT* [53,54,55] and more sophisticated EIT* [56] algorithms. They simultaneously leverage the advantages of both types of algorithms while ensuring asymptotic optimality of the solution, and are among the most computationally efficient random-sampling-based planning algorithms.

Series of RRT algorithms can be combined with other types of algorithms in autonomous vehicle trajectory planning, like cubic B-spline [57], dynamic window approach [58], and biased sampling distribution generated by neural networks with attention mechanism [59].

While random-sampling-based algorithms possess these advantages, the implementations based on random sampling introduce uncertainty into the final planning results. In trajectory planning tasks such as closed-road scenarios where environmental states are relatively certain and there is a high requirement for the quality of the planned trajectory, deterministic trajectory planning results are typically expected. Therefore, the application of such algorithms is not as widespread as graph-search-based, interpolating curve, and numeric optimization algorithms.

The random-sampling-based algorithms discussed and analyzed above are summarized in Table 2.

### 2.3. Interpolating Curves and Finite-Sampling-Based Trajectory Planning

Such algorithms generate interpolated trajectories based on control parameters. Reeds–Shepp curves [60] are composed of a series of straight lines and circular arcs, describing forward straight-line, backward straight-line, and constant curvature turning movements of vehicles. Dubins curves [61] are a simplification of Reeds–Shepp curves and can only describe forward and backward straight-line movements. These parameter curves composed of straight lines and circular arcs are suitable for planning the shortest path under constraints such as curvature and start/end position and heading angle, commonly used in low-speed scenarios such as automated parking. Their disadvantage is the inability to guarantee the curvature continuity of the planned result, making them unsuitable for scenarios with higher vehicle speeds. Clothoid curves (or Euler spirals) [62] have a continuous curvature characteristic that linearly changes with the length of the curve. This characteristic is crucial for limiting the lateral acceleration of vehicles and ensuring comfort while traveling along the trajectory, making clothoid curves widely used in various trajectory planning tasks [63,64]. Their disadvantage is the lack of an explicit analytical mathematical expression. Curve construction relies on integral operations and iterative solving, which are relatively time-consuming [65]. Third-order spirals are also a common mathematical form of trajectory planning curves [66], requiring numerical optimization methods to solve unknown parameters. S-curves (Sigmoid) are applied to simple lane-changing trajectory planning tasks in conventional closed-road scenarios [67]. When considering more starting positions and state constraints such as speed and acceleration constraints, polynomial curves [68,69] are more widely used. Polynomial curves have an explicit mathematical form and can ensure the continuity of speed and curvature changes in the planned result.

In addition to the simple interpolating curves mentioned above, spline curves with piece-wise expressions and multiple control points are also commonly used in trajectory planning algorithms. Polynomial spline curves are one of the most common mathematical representations of planned trajectories. Compared to higher-order polynomial curves, they can use fewer parameters to fit longer distance ranges and more complex geometric shapes of vehicle motion trajectories, and are also more calculation-efficient [70,71]. When using fifth- or seventh-order polynomial spline curves, it is possible to minimize trajectory jerk or the third derivative of acceleration (snap) under given control points, which are related to comfort or actuator energy consumption [72,73]. Bézier spline curves, a special form of polynomial spline curves, are also applied in trajectory planning. They provide control points existing in the form of convex hull points to directly and explicitly determine curve parameters, ensuring that the actual curve is enveloped within the convex hull coverage range. This feature allows for the indirect optimization of continuous Bézier spline curves themselves by optimizing the discrete convex hull of Bézier spline curves through optimization [74,75].

Interpolating curves can be combined with other types of trajectory planning algorithms such as graph-search-based collocation numerical optimization algorithms.

These curves can also function independently. Trajectory planning algorithms that generate predetermined numbers of curve samples are referred to as finite-sampling-based trajectory planning algorithms, as illustrated in Figure 4. Typically, they employ either a uniform objective function [76] or the TOPSIS algorithm [77] for sample selection. These finite samplings occur within action spaces or state spaces. For instance, the dynamic window approach samples within the action space defined by velocity, acceleration, and steering angle [78]. Meanwhile, the lattice planner samples within the state space comprising vehicle lateral positions and longitudinal velocities in the Frenét coordinates [79]. While finite-sampling-based algorithms have simple mathematical forms and high computational efficiency, they may struggle to navigate complex dynamic environments and address complex driving goals over extended timeframes.

The algorithms using interpolating curves discussed and analyzed above are summarized in Table 3.

### 2.4. Shooting Numeric Optimization Trajectory Planning

The shooting methods form optimal control problems (OCPs) for trajectory planning tasks, resulting in sequences of the vehicle’s actions and motion states over time. They typically involve solving an open-loop optimal control problem over a fixed future time interval using discrete time steps, as illustrated in Figure 5. Common types include iterative linear quadratic regulator (iLQR) [80], differential dynamic programming (DDP) [81], and model predictive control (MPC) [15]. These algorithms are not only used for trajectory planning but also for tracking and executing planned results. The OPTPAP algorithm [82] is also a typical algorithm based on OCP formulations.

The MPC algorithms utilizes either nonlinear or linearized vehicle kinematic or dynamic models to predict the vehicle’s state during the planning process, effectively considering complex constraints such as vehicle dynamic constraints. The specific implementation details of MPC algorithms are flexible and diverse. Although the nonlinear MPC (NMPC) algorithm [83], which directly considers nonlinear model constraints, accurately predicts system state changes, it requires significant computational resources. To ensure real-time performance in trajectory planning and tracking tasks, the model needs to be linearized. Specific approaches include the use of linear time-invariant MPC (LTI-MPC) algorithm [84], which directly adopts equivalent linear models based on the flatness properties of vehicle dynamics, linear time-variant MPC (LTV-MPC) algorithm [85], which linearizes the nonlinear model at each control output, and linear parameter-varying MPC (LPV-MPC) algorithm [86], which utilizes linear models in the state space but nonlinear models in the parameter space. Additionally, Hybrid MPC (HMPC) algorithms [87], which switch between different types of models based on rules, and neural network MPC (NNMPC) algorithms [88], which use neural networks as predictive models, also ensure real-time performance. MPC offers flexibility in setting optimization objectives and constraints, with many MPC algorithms incorporating the artificial potential field method discussed in Section 3.4 as part of the risk optimization objective [89,90,91]. Research [92] employs inverse optimal control (IOC) to learn optimal control objectives based on specific task requirements. Planning control algorithms combining MPC with machine learning methods are gaining attention [16].

The iLQR algorithms, compared to the MPC algorithms, usually have more concise mathematical forms. However, their drawback is not explicitly considering complex vehicle dynamics and traffic environment constraints like the MPC algorithms do. Instead, these constraints often need to be added to the optimization objective function through penalty functions [80,93] or augmented Lagrangian methods [94] for iterative solution solving. Additionally, iLQR can only handle linearized state models. On the other hand, iLQR algorithms cannot accelerate the solution process by adopting control time windows inconsistent with the prediction time window as MPC algorithms do. They must solve the entire control sequence within the prediction time window. Despite these limitations, the concise mathematical forms of iLQR algorithms lead to higher computational efficiency [95] and have practical applications in various trajectory planning tasks.

Some newer MPC and iLQR trajectory planning algorithms have taken it a step further by addressing the issue of uncertainty in the behavior of surrounding vehicles. They incorporate the multimodal nature of future trajectories of these vehicles into the optimization process of the algorithms [96,97,98], which provide planning results considering the probability distribution in prediction results. This function also names contingency planning [99]. These algorithms rely on the perception and prediction results of the surrounding vehicles considering uncertainty. Utilizing the multimodal predicted trajectories, these algorithms construct a tree of probable trajectories for the surrounding vehicles, illustrating their behavioral uncertainties. Leveraging the iterative optimization characteristic of MPC and iLQR algorithms, they then determine the optimal action sequence while accounting for these uncertainties.

The shooting numeric optimization planning algorithms discussed and analyzed above are summarized in Table 4.

### 2.5. Collocation Numeric Optimization Trajectory Planning

Unlike shooting methods, collocation methods often do not directly compute specific sequences of vehicle actions and states. Instead, they directly optimize the parameterized trajectories representing vehicle motion, as illustrated in Figure 6. Throughout the optimization process, constraints are applied to the trajectories to ensure their feasibility. Series of discrete control points determines the shape of trajectories.

Collocation methods often employ trajectory representations such as piece-wise polynomial trajectories or polynomial splines [100]. This is because solutions that adhere to vehicle dynamic constraints in shooting methods can be well-approximated in polynomial form at discrete control points using methods like Hermite–Simpson [14]. Piece-wise clothoid curves connecting control points have also been used in trajectory planning with collocation methods [101].

Collocation methods use sparse and discrete control points instead of the complete control sequence solved in shooting methods, generally formed into nonlinear programming (NLP) problems. Polynomial trajectories in them, having explicit expressions and favorable mathematical properties, allow for direct calculation of gradients of various complex optimization objectives with respect to control point parameters in trajectory planning problems. This allows trajectory planning problems with collocation methods to be solved using general gradient-based nonlinear optimization solvers. Representative examples include the general planning algorithm GCOPTER [102] and the efficient spatio-temporal planning algorithm DFTPAV [103]. The latter achieves more efficient computations compared with many famous shooting optimization planning algorithms while considering complex constraints such as vehicle dynamic constraints, static and dynamic obstacle constraints involving geometric shapes, trajectory high-order continuity, and optimization objectives.

When optimization objectives are expressed in concise quadratic forms based on control point parameters, collocation trajectory planning tasks can be reformulated as quadratic programming (QP) problems. Due to the convexity of these problems, the solutions are guaranteed to be unique. Quadratic programming solvers are not only diverse and mature but also more computationally efficient compared to general nonlinear optimization solvers, making QP methods widely adopted in trajectory planning. Specific applications include smooth multi-objective trajectory tracking [71], highway lane changing [104], and minimum curvature trajectory planning for racing [73]. The DL-IAPS+PJSO algorithm [105] formulates different quadratic programming problems based on the distinct characteristics of lateral planning and longitudinal planning, exhibiting high robustness across various scenarios, including parking situations.

Some collocation-based trajectory planning methods do not rely on smooth continuous polynomial trajectories. The widely used Timed Elastic Band (TEB) algorithm employs line segments and circular arcs to connect discrete control points, solving for the shortest-time trajectory while satisfying constraints such as minimum radius, maximum velocity, maximum angular velocity, and maximum acceleration [106]. Some collocation-based methods directly build optimization objectives based on the relative positions between discrete control points, without concern for specific trajectory forms or adopting piece-wise linear trajectories. They approximate real constraints or optimization objectives related to velocity, acceleration, heading angle, and curvature by these discrete points [107,108].

In addition to conventional nonlinear and quadratic programming algorithms, some intelligent optimization algorithms such as particle swarm optimization have also been applied to trajectory optimization for such problems [109].

The collocation numeric optimization planning algorithms discussed and analyzed above are summarized in Table 5.

### 2.6. End-to-End Data-Driven Trajectory Planning

With the rapid development of computer hardware performance and the continuous advancement of artificial deep neural network technology, end-to-end trajectory planning methods are gradually demonstrating immense potential. Existing end-to-end trajectory planning algorithms can mainly be categorized into two types: imitation learning (IL) and reinforcement learning (RL) [17].

IL learns the automated driving planning strategy directly from expert demonstration data, without relying on feedback from interaction with the actual environment. Its objective is to make the behavior of the trained model as close as possible to that of the driver data. IL can be further subdivided into three categories: behavior cloning (BC), direct policy learning (DPL), and inverse reinforcement learning (IRL). BC is a passive offline planning strategy learning method that fits driver data to an end-to-end model and then generates behavior that mimics human driving. BC is the most widely used IL method [18,110]. DPL, based on BC, is an iterative online policy learning method. It continuously improves the policy by effectively eliminating past incorrect strategies [111,112]. IRL, on the other hand, processes expert trajectory data to learn the underlying factors from input to output and optimizes actual trajectory planning strategies based on inferred reward functions. IRL is typically implemented using algorithms such as the maximum margin algorithm [113], Bayesian methods [114], and maximum entropy algorithm [115], flexibly approximating driver behavior based on known data.

Unlike IL, RL relies on interaction with the driving environment to obtain reward and penalty feedback data needed for learning and training. The ultimate goal of RL algorithms is to learn a planning strategy to maximize the feedback accumulated over time. Common RL planning algorithms include deep Q-networks (DQN) [116], deep deterministic policy gradient (DDPG) [117], and proximal policy optimization (PPO) [118,119]. Although RL algorithms typically require plenty of training steps and computational resources and have low sampling efficiency and slow convergence, they can learn complex planning strategies and continually adapt to changing environments. RL algorithms that utilize driver operation guidance information [120], as well as algorithms that combine IL with RL [121], can improve the learning efficiency of RL algorithms.

### 2.7. Spatial and Temporal Spaces, Initial Conditions, and Constraints in Trajectory Planning Algorithms

Different types of trajectory planning algorithms require specific designs of trajectory planning action spaces, state spaces, and constraints, which determine the computational efficiency and quality of trajectory results.

The current hierarchical planning algorithms often project the three-dimensional spatio-temporal trajectory planning problem onto lower-dimensional state spaces in a hierarchical and decoupled manner to reduce computational complexity, solving path planning and speed planning problems sequentially. The decoupled problems not only have smaller state spaces but also have simpler constraint and optimization objective forms, facilitating rapid solutions of complex trajectory planning problems [105,108]. Methods for planning speed curves that avoid obstacles given already planned spatial paths include forward–backward rule-based solvers [31,73] and S–T graph methods [104,108,122].

Meanwhile, some trajectory planning algorithms aim to plan trajectories with paths and speeds simultaneously in a joint time–space domain. This approach, by fully considering the complex interrelationships between vehicle longitudinal and lateral motion characteristics and the detailed patterns of the traffic environment, can yield trajectory planning results that better meet desired objectives. To achieve joint time–space planning in graph-search-based algorithms, appropriate spatio-temporal lattice sampling methods need to be set [25,29,30,123]. Considering that the search process in the spatio-temporal high-dimensional space may result in an exponential increase in the number of samples, some graph-search-based algorithm designs use acceleration interval constraints instead of fixed step-length discrete sampling in the time dimension to effectively avoid the problem of dimension explosion [25,30]. The collocation of numerical optimization algorithms for solving joint time–space problems is reflected in using either the time of discrete control points [106] or the two-dimensional spatial positions at fixed time points [73] as direct optimization objectives, or directly adopting trajectory mathematical forms with time as the independent variable [103].

Apart from QP problems, using shooting or collocation numerical optimization algorithms to solve trajectories may face challenges due to non-convexity and the influence of unexpected local optimal solutions. The most common approach is to integrate numerical optimization methods with other types of trajectory planning algorithms to form hierarchical trajectory planning algorithms, allowing numerical optimization methods to obtain appropriate initial solution conditions. The simplest way is to generate initial solutions based on fixed rules [71,124]. Using graph search methods to generate them is also a common practice. Initial trajectories obtained through graph search can be achieved using Hybrid A* algorithm and its variants in open environments [82,105,125], or by searching for discrete road grid nodes in closed-road scenarios [107,108]. In some algorithm implementations, upper-level graph search trajectory planning algorithms not only provide initial trajectory solutions but also further transform non-convex and redundant obstacle constraints into convex forms suitable for numerical optimization problems based on the initial solution. A common form of constraint transformation is different types of driving corridors [82,107]. Rule-based depth-first search and other methods are also used to generate such corridors [126].

Based on all the discussions in Section 2, the properties of these autonomous vehicle trajectory planning algorithm categories can be concluded and compared briefly in Table 6. The performance of trajectory planning algorithms is closely related to the application scenario, the specific implementation form, and the details of the algorithm. The following conclusions are qualitative analyses and should not be considered definitive.

## 3. Traffic Environment Collision Risk Estimations for Trajectory Planning Tasks

As autonomous vehicles navigate, they encounter plenty of obstacles within their environment, ranging from various traffic participants and stationary objects to the geometry of the road itself. These obstacles form the navigable areas for autonomous vehicles and pose potential collision risks during transit. Equipped with a diversity of sensors, such as cameras, LiDAR, and millimeter-wave radar, autonomous vehicles can perceive different types of obstacles. Autonomous driving trajectory planning algorithms are imperative to calculate a passable path that circumvents the potential collision hazards based on the current perceptual data. However, raw sensory information often contains redundancies and lacks direct representations of actual potential risk states. Depending on the characteristics of different types of trajectory planning algorithms, it is essential to transform, fuse, and process these raw sensor data into formats directly applicable to trajectory planning tasks. A concrete example is illustrated in Figure 7, where the original multi-view images from cameras have been converted into a mathematical representation of 3D bounding boxes describing the poses and geometries of surrounding obstacles, thereby constraining the state space for the trajectory planning algorithms.

In this section, the methods including surrogate safety measures, parallel safety checkers, bounding boxes, occupancy grids, reachable sets, corridors, and different artificial fields that work parallelly with or are directly integrated into the trajectory planning algorithms of autonomous vehicles are compared and analyzed.

### 3.1. Surrogate Safety Measures under Specific Conditions

The measures to estimate the potential risks in the driving environment are named traffic conflict-based measures [128] or surrogate safety measures [129] in transportation research. Parts of them are also applied in autonomous driving techniques, especially in advanced driver assistance systems. Measures like time headway (THW) and time-to-collision (TTC) can not only constitute gap and velocity control strategies in adaptive cruise control (ACC) systems [130] but also describe the velocity control behaviors of human drivers and realize personalized strategies in ACC systems [131,132]. Additionally, some studies have found through fixed-base driving simulator experiments and questionnaires that TTC and THW can describe the subjective perception of the risk changes of the lead vehicle in vehicle following scenarios, which formulates some risk prediction models of human drivers [133,134,135,136]. Similar findings based on THW and TTC measures are also extended to other scenarios like lane changes [137]. Different types of time-to-lane-crossing (TLC) metrics are utilized to describe the subjectively perceived risks in curve negotiation scenarios [138,139].

Apart from applications in ADAS, measures like TTC are also utilized in higher-layer decision-making algorithms, which determine the objectives of trajectory planning algorithms. The typical applications are game theoretic decision-making algorithms in lane-changing [140] and intersection [141] scenarios. Considering that the discontinuity of TTC may lead to unrealistic Nash equilibrium in a game, game theoretic algorithms also use acceleration and time measures based on specific assumptions about traffic participants’ behavior to enhance the decision-making performance [142,143].

Despite their ability to describe traffic risk and drivers’ subjective perception in various scenarios, their mathematical forms may lead to undefined or sudden changes in the calculation results under specific conditions. For instance, it was found that the TTC and TLC indicators produced unexpected jumps and could not accurately measure the risks in lane deviation conditions [144]. Moreover, these indicators can only describe specific risk information in complex traffic environments, and they have shortcomings in different performance dimensions [145], thus limiting their application scenarios. Some studies have attempted to use combinations of multiple measures to alleviate these issues and achieve better estimates of traffic risk, whose effectiveness is relatively limited, leaving much room for improvement [128]. Reliable measures are lacking for the lateral traffic risk in scenarios such as lane changes or merges and the general risks in more complex traffic environments [129]. Accordingly, some emerging risk assessment measures based on the above-mentioned simple surrogate safety measures like 2D TTC managed to consider lateral and longitudinal collision risks simultaneously [146,147].

### 3.2. Safety Checkers Working in Parallel with Trajectory Planning Algorithms

There are comprehensive safety checkers that are applicable to more general and complex driving environments compared with simple surrogate safety measures. At present, they are often used as independent modules to verify the feasibility of planning results.

The Responsibility Sensitive Safety (RSS) model proposed by Mobileye assumes that vehicle drivers need to pay attention to the impact caused by the uncertainty of surrounding traffic participants’ behaviors and uses rule-based mathematical formulas to implement transparent and verifiable real-time traffic risk assessment methods for autonomous driving [148]. These rules describe common-sense behavioral characteristics in human safety driving concepts, essentially considering the thresholds for lateral and longitudinal collision times and safe distances in a comprehensive manner. Specific assumptions such as acceleration thresholds are adopted in the rules, ensuring that the mathematical formulas of these rules correspond closely to the human understanding of common sense [148]. RSS makes autonomous vehicles cautious enough to avoid becoming the cause of traffic accidents as much as possible, and is integrated in Carla 0.9.14 software for autonomous vehicle algorithms simulation [149]. Similarly, Nvidia proposes the Safety Force Field (SFF) model as another set of risk assessment methods for planning control behaviors [150]. Unlike RSS, SFF judges risks based on the intersection of the trajectories of different traffic participants and no longer aims to increase the size of potential risks as the ultimate goal. Although there are specific mathematical differences between RSS and SFF, they have high conceptual similarity and can be used interchangeably, showing similar performance in the Carla simulation environment [151].

However, studies such as [152] argue that the approach of assuming that all traffic participants always follow the same common-sense rules and defining the reasonable behavior of autonomous vehicles based on safe distances, as adopted by RSS and similar methods, is unreasonable. Because even if the behavior of autonomous vehicles themselves is reasonable, other traffic participants may still exhibit behavior inconsistent with the expectations [152]. Therefore, this study proposes to address this issue by considering all the legal behaviors of other traffic participants.

Apart from the rule-based parallel safety checkers discussed above, there are other safety checkers based on the maneuver-oriented motion prediction of other traffic participants. Most of these are data-driven approaches utilizing machine-learning algorithms. For example, structured Bayesian networks trained on datasets can infer collision probabilities based on the predicted future states of the traffic participants, with these predicted future states obtained from Kalman filter models [153]. Future collision probabilities can also be quantified by heuristic Monte Carlo sampling, with the future states estimated from extended Kalman filtering models [154]. Furthermore, the emerging diverse data-driven end-to-end traffic motion prediction approaches provide more reliable future state predictions over longer time ranges compared to the traditional Kalman filter models [155]. The prediction results for the concerned traffic participants in these approaches are in the form of behavioral intention classifications, unimodal trajectories, or multimodal trajectories [155]. Specifically, the safety checkers based on long-short-term memory networks [156] and graph neural networks [157] provide promising risk estimation results based on the predicted trajectories.

### 3.3. Bounding Boxes, Occupancy Grids, Reachable Sets, and Driving Corridors

The most common method of integrating traffic risk modeling into autonomous driving trajectory planning algorithms is the description of whether vehicles can pass through a specific area or avoid collisions, including bounding boxes, reachable sets, and driving corridors. These methods are only used to delineate the boundaries of the passable area for vehicles themselves, without considering the different levels of potential risk at different positions within the passable area based on the motion state of the vehicles.

The trajectory planning methods in structured roads typically employ passive reactive collision check methods, which do not explicitly describe passable areas but instead perform collision checks during the trajectory planning process to determine if the planned trajectory is feasible [76,79,158]. Collision checks are usually based on the geometric bounding boxes of the vehicle and other traffic participants [74,77,108], as illustrated in Figure 8. In specific scenarios such as lane changing, collision detection can be further simplified to one-dimensional distance constraints [159].

With the development of advanced end-to-end autonomous vehicle perception and prediction algorithms, three-dimensional occupancy grids become promising substitutes for the traditional bounding boxes. They provide detailed spatial occupancy information of the obstacles in the environment beyond simple bounding size and position information, providing finer geometry details and possessing privileges in describing out-of-vocabulary objects [160]. Currently, the most popular and efficient realizations are based on emerging Transformer networks [161].

Reachable sets encompass the collection of states that autonomous driving vehicles can reach over time starting from a series of initial states without collisions, and they have been widely applied in planning algorithms [162,163]. Tools like SPOT have been developed for generating reachable sets [164]. Improved reachable sets can handle environments of arbitrary complexity [163]. Apart from the motion dynamics of transportation participants, some reachable set designs also consider the visibility of the areas around the driver’s field of vision [165,166]. The reachable set considering the above-mentioned two types of environment information is illustrated in Figure 9. The reachable sets mentioned above are suitable for conducting reachability analysis in complex traffic environments, generating passable areas for autonomous vehicles, effectively exploring the available state spaces and the topological structure for driving tasks, even detecting narrow passages.

Similar to the concept of reachable sets are driving corridors. Driving corridors are typically generated based on the initial reference trajectory and environmental obstacle constraints. Some examples of convex driving corridors and their relationships with planning algorithms are provided in Section 2.7. Driving corridors transform the original complex and multi-source constraint information into convex constraints for numeric optimizations [103,126], as illustrated in Figure 10. They can be considered as the collection of feasible motion states provided by the high-level decision module or based on reachable sets, simplifying the trajectory planning problem. If no feasible driving corridor exists in a certain area, all the planning results in that area are considered unreasonable [167].

Bounding boxes, reachable sets, and driving corridors provide similar information about passable areas. Therefore, some new planning algorithms propose to uniformly handle three different types of risk constraints in optimization objectives, ensuring the compatibility of planning algorithms with different types of upstream perception and prediction information [98]. Although the traffic risk considerations applied to trajectory planning algorithms effectively describe information about passable areas in the environment to avoid collisions, they typically assume that the predicted trajectories are stably executed and cannot describe the potential risk levels caused by changes in the relative position and relative velocity. Although some planning algorithms introduce the Euclidean distance between the vehicle and the boundaries modeled by the three methods into the optimization objectives [98,168], which can introduce the modeling of potential risk levels to some extent, they still overlook the nonholonomic dynamic properties of vehicles and ignore the differences in the potential risk characteristics in the lateral and longitudinal directions of vehicles.

### 3.4. Potential Fields, Virtual Force Fields, and Composite Risk Fields

Various virtual field methods have been widely used not only to achieve obstacle avoidance but also to measure the potential risks in the driving environment. These fields can be based on the concrete artificial potential fields representing different traffic participants and relative movements between roads in the tangible traffic environment [169], or they can be abstract virtual force fields based on temporal distances beyond traditional spatial distances in the state space during trajectory planning [35].

The traditional artificial potential field methods establish a single potential field in the traffic environment, as illustrated in Figure 11. The typical designs often include a virtual repulsive force that decays with the Euclidean distance to obstacles, penalizing close relative positional relationships during trajectory planning [89,170]. To further consider the risk impact caused by relative motion, some improved artificial potential field methods have potential function distributions that change with the relative velocity between the ego vehicle and obstacles [91,171,172]. This design is similar to considering the minimum safe distance with respect to the relative velocity but extends it to two-dimensional space rather than being limited to one-dimensional longitudinal risk considerations in specific scenarios. Some artificial potential field designs further consider the impact of relative acceleration [169]. Although these artificial potential field designs can achieve safety goals such as minimum safe distance, they often do not represent the characteristics of real traffic collision risks and do not provide clear physical interpretations of potential collision risks like safety surrogate metrics such as time headway (THW), time to collision (TTC), and post-encroachment time (PET) [129]. On the other hand, the existing artificial potential field methods provide almost no quantitative parameter calibration methods to accurately measure the traffic risk conditions. Only a small amount of literature provides qualitative analyses of parameter calibration methods [171]. Despite the shortcomings of the existing artificial potential field methods, they provide a unified approach for modeling different sources of risks in complex traffic environments and can consider the potential risks brought by relative positions and relative velocities before real collisions happen. The relationship between artificial potential field designs and the desired risk goals in trajectory planning algorithms requires further exploration.

In recent years, the introduction of composite risk field methods has made it possible to address the issues with single artificial potential fields, achieving more precise and detailed modeling of traffic risks. The critical idea is to first model the influences or risk consequences of obstacles in the traffic environment using one artificial field, and then model the probability of future events in the traffic environment using another artificial field. The final risk estimation result therefore comes from the interactions between these two fields, as illustrated in Figure 12. Probabilistic Driving Risk Field (PDRF) assumed the probability distribution of the lateral and longitudinal acceleration magnitude of the ego vehicle over a fixed future time period and modeled the composite risks using the method of the overlap area of the vehicle geometric bounding boxes considering future motion and the probability distributions of the accelerations in this area [144]. Driver’s Risk Field (DRF) is proposed to represent the subjective judgment of the driver on the possible future positions of the ego vehicle within a look-ahead time range interval [173]. The DRF extends along the predicted trajectory of the vehicle and changes its geometric shape based on the ego vehicle’s state, such as velocity and steering wheel angle. The dual integration result of the DRF multiplied by the field representing the influence of obstacles provides the final estimate of the traffic environment risk [173,174]. In addition, related studies have proposed specific procedures and methods for calibrating DRF parameters using driving simulator experimental data [173] or real vehicle experimental data [174]. Both PDRF and DRF rely on two-dimensional integration operations, requiring a considerable amount of computational resources [144,173]. This disadvantage is particularly evident for DRF, which needs to perform two-dimensional integration over large spatial areas in the traffic environment. The two-dimensional integration operations also make it difficult for PDRF and DRF to provide the gradient information of estimated risk magnitude with respect to vehicle states. In contrast, the traditional single artificial potential field methods can obtain this gradient information at a low cost [91], speeding up the convergence of numerical optimization trajectory planning algorithms. The Geometric Driver Risk Field (GDRF) algorithm is based on more efficient geometric intersection calculations rather than two-dimensional integrations, retaining the benefits of the comprehensive risk modeling details inherent in composite risk field methods while significantly enhancing the computational efficiency for real-time trajectory planning tasks. GDRF can also be integrated with gradient-based collocation numeric optimization planning algorithms [175].

The risk estimation methods in trajectory planning discussed and analyzed above are summarized in Table 7.

## 4. Human Driver Behaviors and Expected Autonomous Vehicle Trajectory Planning Behaviors

Various survey-based theoretical models, including the Technology Acceptance Model (TAM), Unified Theory of Acceptance and Use of Technology (UTAUT), Theory of Planned Behavior (TPB), and Innovation Diffusion Theory (IDT), have been employed in previous studies to identify the factors influencing users’ expectations of autonomous vehicles [176]. For instance, studies utilizing TAM have shown that a driver’s trust is crucial in shaping their perception of risks, usefulness, and intention to adopt autonomous vehicles [177]. To enhance driver trust, TAM highlights several key aspects: system transparency requires vehicles to demonstrate consistent and predictable behaviors; technical competence demands high performance with minimal errors across various scenarios; and situation management expects the provision of adequate and responsive alternative solutions under the driver’s control [177]. Achieving these aspects necessitates appropriate actions that align with the driver expectations, which in turn require a thorough understanding of the complex interactions between the traffic participants and the environment. Accordingly, other studies have stated that personalized or human-like driving behaviors in autonomous vehicles have the potential to enhance road safety, transportation efficiency, and human-centric mobility [21].

Companies like Bosch have already introduced commercial solutions for personalized autonomous driving behaviors, such as Dynamic Distance Assist (DDA) [178]. However, the existing solutions can only achieve personalized driving behaviors on relatively basic and well-defined dimensions, such as following distance. The realization of more complex and human-like personalized autonomous driving behaviors remains to be further explored. This requires an in-depth discussion on the utilization of driver behavior data.

In this section, methods including imitation learning, preference learning, and driver behavior modeling that can incorporate the data of human driver behaviors into trajectory planning algorithms are investigated. The critical ideas of the three methods are illustrated in Figure 13. The influences of these methods on users’ subjective expectations are also discussed.

### 4.1. Mimicking Human Trajectories Directly: Imitation Learning and Data-Driven Model Fitting

Section 2.6 has introduced the imitation learning end-to-end trajectory planning algorithms. These end-to-end data-driven methods fully utilize the flexibility and generalization capabilities of advanced deep learning techniques to directly imitate the manual driving trajectory data collected from human drivers.

In addition, some methods adopt model-based data-driven driver trajectory imitation learning approaches. These methods do not directly learn driver trajectory data themselves but instead establish parameterized models combined with other trajectory planning algorithms representing different dimensions of driver behavior properties [20], naming implicit personalization models. The most common models include Hidden Markov Models (HMM) and Gaussian Mixture Models (GMM), either individually or in combination, which have been found through real vehicle validation to fit real longitudinal and lateral driving behavior data [179,180]. Maximum Likelihood Estimation (MLE) [181], Recursive Least-Squares (RLS) [182], and logistic regression models [183] have also been applied to personalized model establishment. The study in [184] not only employs the Nelder–Mead simplex optimization method for the offline identification of driver steering model parameters but also utilizes the unscented Kalman filter (UKF) for online real-time parameter learning. Unlike imitation learning methods, parameterized models are typically limited to more specific driving tasks rather than general autonomous driving planning tasks. However, these models’ methods offer better interpretability, directly representing the characteristics of different driver behaviors clearly. These model-based methods are reported to not only improve the driver acceptance of the system under specific conditions but also reduce the false alarm rates of ADAS systems [180].

Other popular methods include the identification and classification of the driving styles of human drivers [185,186] and designing distinguished trajectory planning strategies for different styles accordingly [90].

Although the methods mentioned above, which directly mimic human driver trajectories or driving styles, show promising results, there is still controversy surrounding their effectiveness. Some studies have indicated that drivers’ preferences and expectations of autonomous vehicle behaviors differ from their own driving behaviors observed in the collected manual driving trajectory data across diverse driving scenarios [187,188,189], presenting a challenge to these mimicking methods. These differences have been found to be associated with specific driving scenarios [188]. For instance, the relationship between personalized speed control and subjectively perceived safety is moderated by trust in autonomous vehicles. Personalized speed control behavior only positively influences drivers’ subjectively perceived safety when they lack trust in the autonomous driving system [190]. Therefore, it is imperative for future research to further explore and determine the specific autonomous driving scenarios and the specific user groups for which these mimicking methods are suitable.

**Figure 13 sensors-24-04808-f013:**
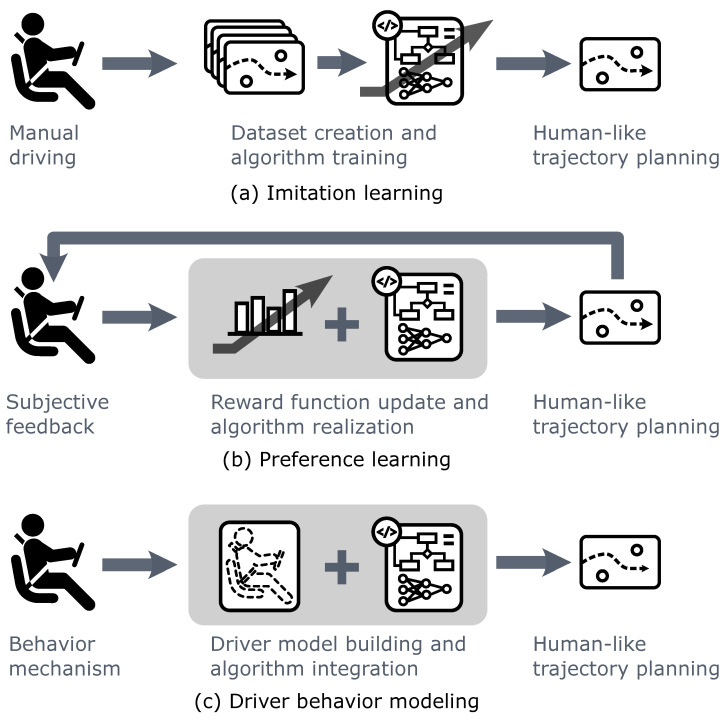
The illustrations of methods incorporating data of human driver behaviors into trajectory planning algorithms, including imitation learning, preference learning, and driver behavior modeling.

### 4.2. Learning from Human Feedback: Preference Learning

Different from those methods discussed in Section 4.1, some studies aim to learn drivers’ experience and preference knowledge through direct feedback information from drivers rather than indirectly collected manual driving data.

Preference-based learning methods have been widely used in the field of robotics to help complete complex operational tasks without explicitly determining the reward functions and their parameters. These methods utilize the preference information from human users about specific trajectories to assist in task completion. Similar to imitation learning, preference-based learning can either directly learn the optimal strategy itself [191] or learn a lower-dimensional and more interpretable reward function [192]. Such methods have also been applied to learning the driver behavior preferences in automated driving vehicles [193,194]. The reward function is typically a linear combination of the parameter set [195]. Research has also used neural networks to learn the nonlinear relationship between the size of the reward function and the parameters of the reward function [194]. Another study further combined preference-based learning with imitation learning of user-demonstrated trajectories to achieve a hybrid strategy learning method [196].

Preference-based learning demonstrates the potential to learn complex preference behaviors using minimal data. However, in practical applications, it typically requires dozens or even hundreds of inquiries for drivers to repeatedly experience similar driving scenarios to learn the final goal [197]. Some research has further incorporated the subjective feedback acquisition of preference features based on preference inquiries, which improves the convergence speed of the learning process compared to basic preference-based learning methods [198]. In specific tasks like the preference learning of lane centering control (LCC) behaviors [199], the feedback query method and the reward function parameter update policies can be designed in detail to learn personalized preferences within only a dozen inquiries. Therefore, such methods need to be further explored in combination with specific practical application scenarios to explore further application possibilities.

The efficiency issue limits the application of preference learning to autonomous driving tasks. In addition, recent research has also focused on large language models based on Transformer architecture for human preference learning to understand driver feedback information and adjust the behavior of automated driving vehicles accordingly, showing potential applications [200,201].

### 4.3. Utilizing Human Behavior Mechanism: Driver Modeling

Apart from utilizing human trajectory data and human feedback, some research aims to achieve driving behaviors that align with drivers’ expectations through modeling the mechanism behind drivers’ behavior.

Traditional model predictive control (MPC) algorithms, besides optimizing the objectives related to obstacle avoidance, driving efficiency, and smoothness, typically include an optimization objective related to the deviation distance from the lane centerline to steer the trajectory as close to the lane centerline as possible [90,91]. However, the assumption of driving along the lane centerline does not align with real drivers’ driving habits. Studies have found that human drivers typically adjust the vehicle’s lateral position with specific patterns during the different phases of cornering, with trajectories closer to the lane centerline being relatively less common [202,203]. To address this issue, studies from different institutes collected extensive data on the lane-keeping processes from drivers and analyzed the lateral offset patterns in these data. Based on these preference patterns, they dynamically generated driving corridor regions and set lateral position constraints for MPC algorithms, achieving human-like driving trajectories by minimizing the steering wheel manipulation rather than following the lane centerline [204,205]. The dynamically generated driving corridors based on lateral offset patterns can be viewed as a concise and practical method for modeling driver behavior mechanisms. When it comes to the velocity control of vehicles, the intelligent driver model (IDM) and its variants are widely adopted for the longitudinal motion planning of autonomous vehicles [206].

The in-depth research on driver perception and behavior also poses new challenges to the other common assumptions or optimization objectives used in the existing automated driving planning and control algorithms. As early as 1970, researchers found that drivers are not sensitive to the relative distance deviation between two points in their forward view [207,208], challenging the modeling of driver behavior based on the deviation distances of preview points. Studies in psychology have found that drivers cannot accurately estimate the curvature of the road ahead [209]. Research in traffic engineering has found that one of the main reasons for accidents on curved roads is drivers’ incorrect estimation of the curvature of sharp curves, providing evidence for this [210]. Therefore, planning algorithms that assume drivers can estimate the curvature of upcoming curves to adjust their steering behavior may not consider this perceptual regularity of drivers [211]. The parts of planning algorithms with trajectory curvature as the sole optimization objective [158,212] may overlook drivers’ deep-seated psychological expectations and behavior mechanisms.

Furthermore, the research on driver perception and behavior can provide new insights for the algorithm design in automated driving systems. Researchers have found that drivers’ gaze behavior during cornering is directly related to their steering control behavior and the desired driving trajectory [213,214]. Moreover, more research indicates that drivers’ steering control processes mainly rely on visual information [215].

The research on which key features of visual information drivers use to produce corresponding driving behaviors has a history of more than forty years. Due to the relative motion in the environment and the visual persistence effect of drivers, all the visible surrounding objects will leave plenty of radial line-shaped images in the human field of view, known as optic flow. Studies have shown that drivers can use optic flow information to estimate the vehicle speed [216], heading angle [217], and future vehicle motion trajectories [218,219]. Adding optic flow information to flickering visual images can significantly improve drivers’ steering performance, with more pronounced effects than increasing frame rates [215]. This confirms the mechanism of drivers’ predictions of vehicle motion based on optic flow. Experiments have also shown that, during cornering, drivers’ preview gaze positions alternate between closer and farther positions along the road at regular intervals. Drivers’ preview gaze behavior at closer road positions, called guiding fixation (GF), is directly related to drivers’ current steering control objectives [220]. Drivers’ preview gaze behavior at farther road positions, known as look-ahead fixation (LAF), involves drivers planning the future driving trajectories and assessing the future risks [221]. Research found that changes in the angular projection of the front vehicle in the field of view are directly related to drivers’ subjective perceptions of safety margins and collision risks [134].

Applying the findings of the above studies to autonomous vehicle algorithms has been explored preliminarily. The challenge lies in how to transform the diverse perception and behavior patterns discovered into models that can be integrated and applied to trajectory planning algorithms. The typical models that interconnect drivers’ visual gazing behaviors and curve negotiation behaviors include the tangent point models [213,222] and waypoint models [219,221]. However, these models are usually closed-loop reactive control models based on simple state parameters and can only describe the driving behaviors in specific scenarios without fully considering the complex prediction, decision-making, and planning behaviors of real drivers in actual driving tasks. A comprehensive and systematic review [223] suggests that a unified and universally applicable driver behavior model based on control theory, perceptual psychology, and neuroscience is likely to become possible in the near future. The composite risk fields introduced in Section 3.4 try to establish subjective risk models based on drivers’ subjective attentions considering the steering behaviors and preview behaviors in complex environments applicable for more complex planning tasks.

Ongoing studies are trying to mitigate the gap between complex driver behavior modeling and trajectory planning algorithm development. For example, the AV-IDM model is proposed to enhance the traditional longitudinal intelligent driver model for lateral and longitudinal motion planning simultaneously and human-like reactive trajectory generations, with the parameters calibrated with the fractional factorial design approach [224]. The GDRF model is integrated with hierarchical trajectory planning algorithms to realize human-like multi-objective trajectory planning in various driving scenarios [175]. An end-to-end framework PHTPM that mimics the drivers’ preview behaviors around the tangent point and the preview point is proposed for trajectory planning tasks [225]. A linear driver model is also proposed to generate the waypoints ahead, reflecting the human driver preferences for trajectory planning [226].

The methods utilizing human data in trajectory planning discussed and analyzed above are summarized in Table 8.

## 5. Challenges and Future Perspectives

Based on the discussions in the previous sections, the challenges faced by the industry and autonomous vehicles considering trajectory planning tasks can be summarized as follows:Technique Requirements: The continuously evolving market for autonomous passenger vehicles and robotaxis is placing increasingly higher demands on the performance of autonomous driving techniques. Each type of trajectory planning algorithm has its limitations. Even relatively general trajectory planning algorithms may encounter specific issues in certain driving conditions. Ensuring the robustness and high performance of planning algorithms in complex scenarios with uncertainties from various sources poses a significant challenge. However, these efforts are essential for transportation safety and the widespread adoption of autonomous technology.Safety and Regulations: The ongoing popularization of autonomous driving technology faces various legal and regulatory restrictions and encounters complex traffic scenarios where autonomous vehicles, traditional manually driven vehicles, pedestrians, and non-motorized vehicles coexist. Autonomous vehicles need to handle complex safety objectives in real time. The current trajectory planning algorithms primarily utilize fast and direct methods such as bounding boxes, reachable sets, and corridors. Additionally, incorporating considerations for the potential risks arising from environmental uncertainties, driver behaviors, and vehicle dynamics into the algorithms is crucial for ensuring road safety.User Experience and Market Expectations: The user experience of autonomous driving systems is becoming a key factor influencing their market competitiveness and adoption levels. The current manually set objectives and parameters in trajectory planning algorithms may not align with diverse users’ subjective expectations. Methods such as imitation learning, preference learning, and driver behavior modeling have different limitations in addressing this issue. More systematic investigations are needed to develop approaches for designing trajectory planning algorithms oriented towards user experience.

Accordingly, considering the challenges mentioned above, potential future perspectives could be outlined below:Enhanced Objective Integration: Faced with the requirements from planning task constraints and user demands, it is essential to incorporate more comprehensive and detailed objectives concerning various real-world driving tasks into the designs of trajectory planning algorithms. Precisely quantifying these needs through objective metrics and high-quality datasets will make significant differences.Advanced Trajectory Planning: As the application scenarios of autonomous driving systems continue to expand and the requirements for the takeover rates increase, more advanced trajectory planning algorithms capable of handling complex tasks will emerge. For example, adopting spatio-temporal trajectory planning instead of separate path and velocity planning can yield more optimal results in complex scenarios. Additionally, enhancing the consideration of the behavioral uncertainty among the traffic participants based on the available perception and prediction information is a promising development direction.Safety and Risk Estimation: Safety and regulatory concerns call for integrating more advanced, realistic, yet efficient methods for estimating the potential collision risks into planning algorithms to promote safer driving behaviors. For instance, advanced occupancy prediction technologies that provide comprehensive information about the geometries, motions, statuses, and behavioral intentions of traffic participants or other obstacles deserve further investigation. These technologies can be integrated with both end-to-end planning algorithms and traditional modular approaches.Human-like and Interpretable Planning: Techniques that generate more consistent, interpretable, and human-like trajectory planning results will attract more attention. It is critical to cultivate human-like driving behaviors that meet users’ expectations through the systematic development of trajectory planning algorithms and diverse parameter tuning methods. Simultaneously, developing more reliable end-to-end trajectory planning methods based on data-driven approaches, while providing more interactions and feedback information, may alleviate users’ concerns.

## 6. Conclusions

This review paper encompasses numerous studies concerning the behaviors of autonomous vehicles and the realization of trajectory planning tasks, covering three main topics: trajectory planning algorithms, the collision risk estimation methods involved in these algorithms, and approaches for achieving trajectory planning results that align with the user expectations.

In summary, the discussed planning algorithms are categorized into various types, including graph-search-based, random-sampling-based, interpolating curves, shooting optimization, collocation optimization, and end-to-end algorithms. The risk estimation methods encompass surrogate safety measures, parallel safety checkers, bounding boxes, occupancy grids, reachable sets, corridors, and various artificial fields. Additionally, user experience-oriented approaches utilizing human data such as imitation learning, preference learning, and driver behavior modeling are explored.

This paper provides a comprehensive comparative analysis of the algorithms and techniques, uncovering their interconnections between the three topics.

## Figures and Tables

**Figure 1 sensors-24-04808-f001:**
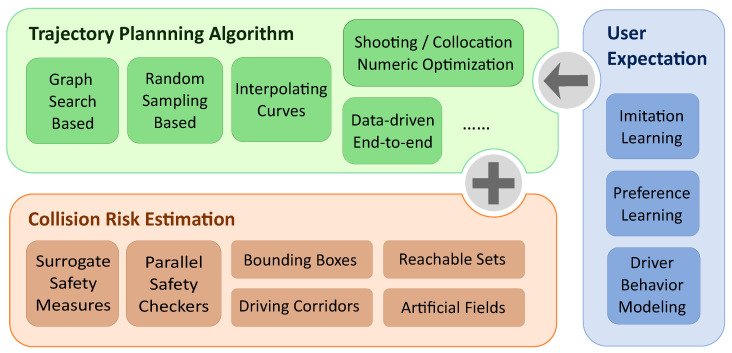
The relationships between trajectory planning algorithms, collision risk estimation methods, and user expectation approaches reviewed in this paper.

**Figure 2 sensors-24-04808-f002:**
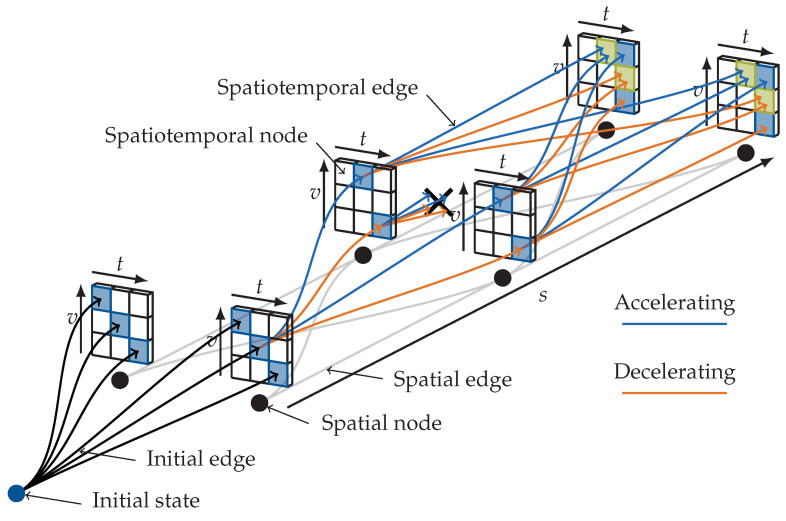
Spatio-temporal graph-search-based trajectory planning algorithm in [25].

**Figure 3 sensors-24-04808-f003:**
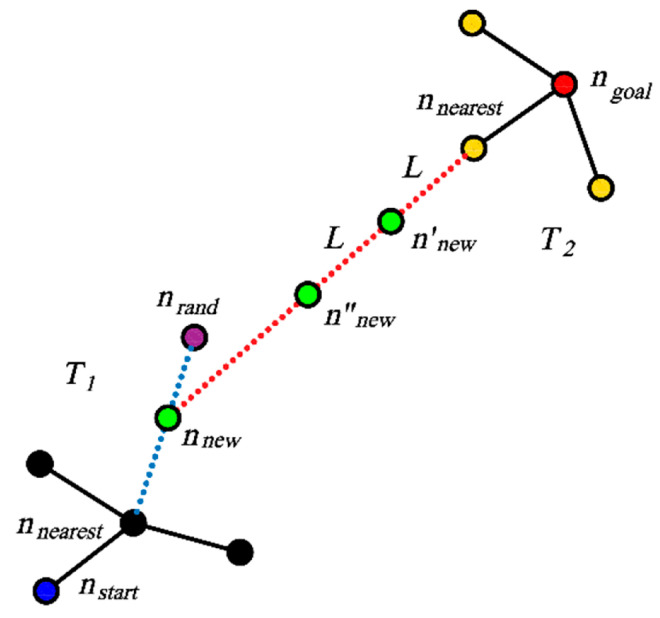
Two-way search tree expansion in Bi-RRT algorithm [46].

**Figure 4 sensors-24-04808-f004:**
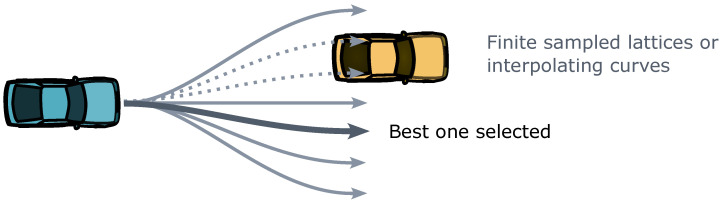
The illustration of finite-sampling-based algorithms based on interpolating curves.

**Figure 5 sensors-24-04808-f005:**
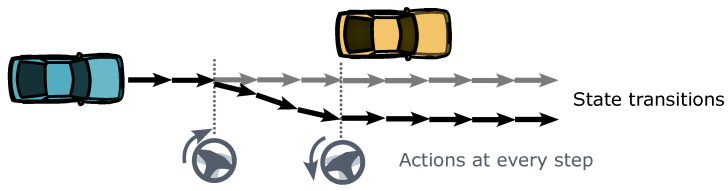
The illustration of shooting optimization trajectory planning algorithms based on optimal control problems.

**Figure 6 sensors-24-04808-f006:**
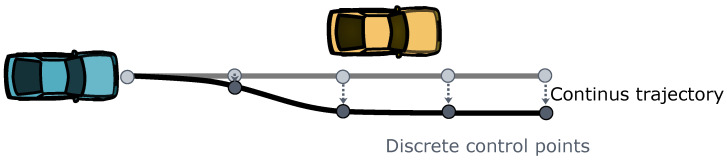
The illustration of collocation optimization trajectory planning algorithms based on control points and parameterized trajectories.

**Figure 7 sensors-24-04808-f007:**
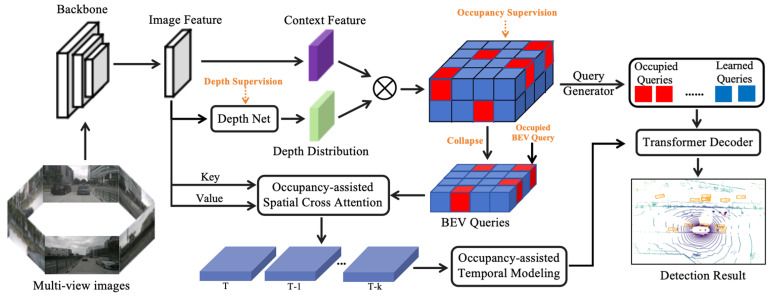
Intuitive example of transformation from raw sensor data (multi-view images) into mathematical formats suitable for trajectory planning algorithms (3D bounding boxes) [127].

**Figure 8 sensors-24-04808-f008:**
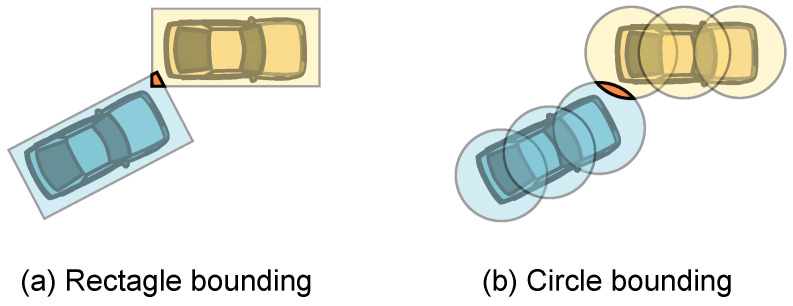
The illustration of typical bounding boxes used in trajectory planning.

**Figure 9 sensors-24-04808-f009:**
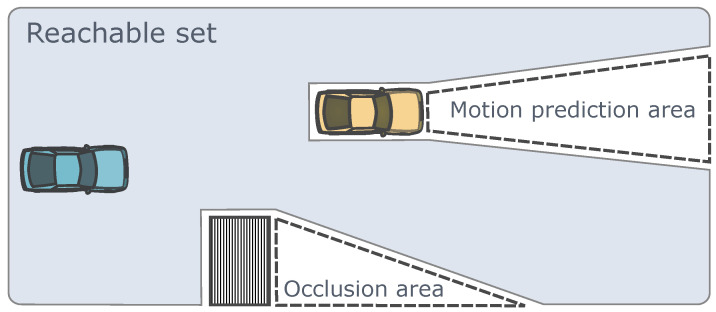
The illustration of reachable set considering the obstacle motion predictions and visibility in the field of vision.

**Figure 10 sensors-24-04808-f010:**
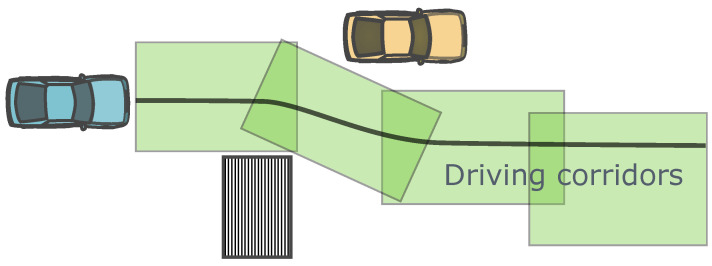
The illustration of driving corridors for convex constraints in the trajectory planning algorithms.

**Figure 11 sensors-24-04808-f011:**
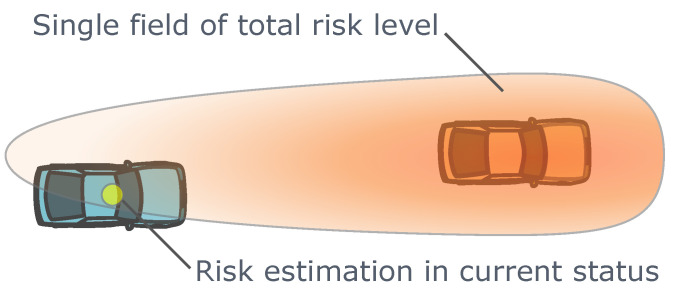
The illustration of traditional artificial potential field methods for risk estimations.

**Figure 12 sensors-24-04808-f012:**
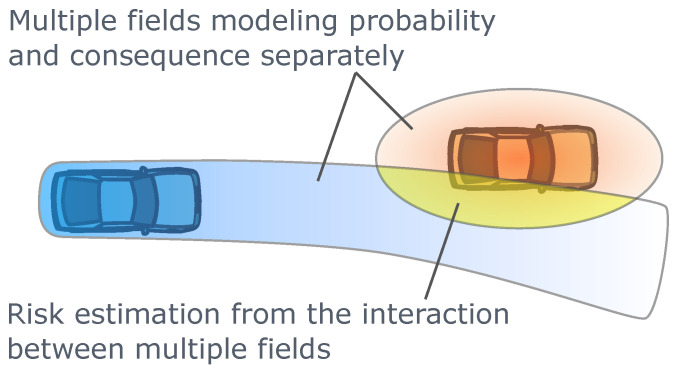
The illustration of composite risk field methods for risk estimations.

**Table 1 sensors-24-04808-t001:** Summary and comparison of graph-search-based planning algorithms.

Algorithms	Pros	Cons
Exhaustive search [30,31], Dijkstra [22,23,24], UCS [25]	Generic, cost function flexibility, optimality in discrete space	No heuristic, relatively low calculation efficiency
A* [26,27,28,29]	Wide applications, calculation efficiency with heuristic, optimality in discrete space	Restricted cost definition brought by heuristics
Hybrid A* [32,33], THA* [34]	Consideration of kinematic constraints, high efficiency brought by space discretization	Suboptimality in discrete space
ATRWA* [36], D* [37], D* Lite [38], LPA* [39], iADA* [40]	Reusing historical planning information, adapting to dynamic environments	Limited research in autonomous vehicle applications currently

**Table 2 sensors-24-04808-t002:** Summary and comparison of random-sampling-based planning algorithms.

Algorithms	Pros	Cons
PRM [41], PRM* [42]	Efficient re-planning in common maps	Time-consuming collision detections
RRT [44]	Probabilistically complete, Generic	No asymptotic optimality
RRT-connect [45]	Probabilistically complete, high computation efficiency	No asymptotic optimality
RRT* [42,47,59], Informed RRT* [48]	Probabilistically complete, asymptotic optimality, wide applications	Improvements needed in application details
Kinodynamic RRT* [49], KB-RRT* [50]	Advanced dynamic constraints compared with RRT*	Low computation efficiency for constraints/Model-driven pruning not reliable enough
FMT* [52], BIT* [53,54,55], EIT* [56]	Probabilistically complete, asymptotic optimality, high efficiency based on graph theory	Limited research in autonomous vehicle applications currently

**Table 3 sensors-24-04808-t003:** Summary and comparison of planning algorithms using interpolating curves.

Algorithms	Pros	Cons
Reeds–Shepp [60], Dubins [61]	Simple and efficient	Curvature discontinuity, low-speed only
Clothoid [63,64]	Continuous and linearly changing curvature	No explicit analytical expressions, low computation efficiency
Third-order spiral [66], S-curve [67]	Suitable for applications in special scenarios	
Polynomial [68,69], Polynomial spline [70,71,72,73], Bézier spline [74,75]	Explicit analytical expressions, high computation efficiency, easy derivatives calculation	Suboptimal trajectory potential considering OCP

**Table 4 sensors-24-04808-t004:** Summary and comparison of shooting numeric optimization planning algorithms.

Algorithms	Pros	Cons
MPC and its multiple variants [83,84,85,86,88,89,90,91,92,96]	Consideration of vehicle dynamics and other constraints, flexible and versatile formulation, inconsistent preview and planning window length	Low computation efficiency, requirement for problem simplification
iLQR [80,93,94,97,98]	Simple problem definitions, high calculation efficiency	Only linear vehicle dynamics, no explicit hard constraints, consistent preview and planning window length
DDP [81]	More accurate vehicle dynamics compared with iLQR	Limited research in autonomous vehicle applications currently

**Table 5 sensors-24-04808-t005:** Summary and comparison of collocation numeric optimization planning algorithms.

Algorithms	Pros	Cons
NLP optimizations [101,102,103,106]	Achieving complex objectives in specific problems while maintaining computational efficiency	Transformations of complex constraints, objectives, and their derivatives required
QP optimizations [71,73,104,105,107,108]	Usually high computation efficiency, single optimal solution guaranteed	Limited objectives in quadratic forms

**Table 6 sensors-24-04808-t006:** Summary and comparison of the properties of different trajectory planning algorithm categories.

Planning Algorithm Category	Pros		Cons		Typical Use
Graph search	High efficiency in exploring non-convex state spaces	Flexible planning objectives, optimal or suboptimal solutions in discrete spaces	Non-smoothness in searched trajectories and need post-processing	Predefined discretization of the state space and action space required	Serve as “front-end” in trajectory planning frameworks
Random sampling		No predefined space discretization needed		Randomness in the planning results	
Interpolating curves, finite sampling	Easy implementations and low computation burden		Hard to cope with complex planning tasks in long time ranges		Basic components of other categories
Shooting numeric optimization	Continuous, smooth, and executable trajectory generation	Direct and adequate incorporation of vehicle dynamic constraints	Typically rely on initial solutions, convex objectives, state and action constraints	Simplification of optimization problems needed or sacrificing efficiency	Serve as “back-end” in trajectory planning frameworks
Collocation numeric optimization		Higher efficiency in realizing multiple planning objectives		Indirect dynamic constraints	
End-to-end data-driven	Flexible data utilization and strategy learning methods.		Black-box nature, difficulty in locating source of mistakes		Paradigm and structures in evolution

**Table 7 sensors-24-04808-t007:** Summary and comparison of risk estimation methods in trajectory planning.

Method	Pros	Cons
Surrogate safety measures [128,129,130,131,132,133,134,135,136,137,138,139,140,141,142,143,145,146,147]	Simple definitions, clear physical significance, high calculation efficiency, subjective perception consistency	Limited in specific scenarios, discontinuity
RSS [148,149], SFF [150,151], other rule-based safety checkers [152]	Generic in complex scenarios, integrated with simulators like CARLA	Not directly integrated with algorithms, unrealistic rule assumptions
data-driven safety checkers [153,154,156,157]	Flexible data fitting ability with good performance	Trained models usually only fit specific driving scenarios
Bounding boxes [74,76,77,79,108,158]	Simple definitions, high computation efficiency	Lack of detailed information for potential risk estimation
Occupancy grids [160,161]	Detailed geometry information for out-of-vocabulary obstacles	Promising but inadequate status information for potential risks currently
Reachable sets [162,163,164,165,166]	Capturing abundant scenario information	Lack of potential risk information, conservative
Driving corridors [103,126,167]	Convex constraints for optimization algorithms	Lack of potential risk information
Traditional artificial potential fields [35,89,91,168,169,170,171,172]	Unified modeling of multiple risk sources, high calculation efficiency, gradient information	deviation for realistic risk status or subjective perception
PDRF [144], DRF [173,174]	Detailed risk modeling based on probability and severity	Low computation efficiency, no gradient information
GDRF [175]	More efficient than DRF while detailed, gradient information, subjective perception consistency	Relative lower computation efficiency

**Table 8 sensors-24-04808-t008:** Summary and comparison of methods utilizing human data in trajectory planning.

Method	Pros	Cons
Imitation learning [18,110,111,112,113,114,115], implicit personalization models [179,180,181,182,183,184], driving style classification [90,186]	Data-driven, flexibility in mimicking human driver trajectory data	Deviation from human preferences in some scenarios, clarification of applicability ranges needed
Preference learning [191,192,193,194,195,196,198,199,200]	Planning algorithm updates from human feedback directly	Large amounts of user interactions required currently, algorithm structure under development
Driver behavior modeling [175,204,205,206,224,225,226]	Realizing human-like behaviors based on behavioral mechanism	Integration methods of driver models and planning algorithms need further investigation

## Data Availability

No new data were created or analyzed in this study.

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
