# Peer review of "A Survey of Autonomous Vehicle Behaviors: Trajectory Planning Algorithms, Sensed Collision Risks, and User Expectations"

_sensors, 2024, doi:10.3390/s24154808_

Round 1

Reviewer 1 Report

Comments and Suggestions for Authors

This article has conducted a thorough investigation in Trajectory Planning Algorithms, Sensed Collision Risks and User Expectations for autonomous vehicle, and provided a detailed analysis. Following suggestions can be considered.

1. Section 2 mainly addresses the local Trajectory Planning problem, can the authors provide more analysis on global trajectory planning?

2. The article extensively analyzed different methods, but did not provide any simulation or experimental results. Can the authors provide some simulation results to enhance the readability of the article?

3. Automatic vehicles can work in many spaces, including land, underwater, air, and so on. Are these methods investigated in this article universally applicable? 

Comments on the Quality of English Language

The overall language expression of the article is good.

Author Response

[Please refer to the PDF version of response letter for the complete content]

Comments 1: Section 2 mainly addresses the local Trajectory Planning problem, can the authors provide more analysis on global trajectory planning?
Response 1:

Thanks for stressing the difference between local trajectory planning algorithms and global trajectory / navigation planning algorithms. We further differentiate between them and provide more analysis in Section 2 of the revised manuscript.
```
It is worth noting that the graph-search-based and random-sampling-based algorithms discussed below are also widely applied to global path planning tasks in autonomous driving navigation tasks. However, this article primarily focuses on their applications in collision-free local trajectory planning tasks. These tasks can be accomplished using similar algorithms but with different problem definitions.
```

Comments 2: The article extensively analyzed different methods, but did not provide any simulation or experimental results. Can the authors provide some simulation results to enhance the readability of the article?
Response 2:

Thanks for your concerns about readability. However, we believe it is not necessary to add simulation results for the following three reasons:
- This is a review article, and it is uncommon to include simulation results in such articles. The existing relevant reviews do not include them either [citations in the PDF version].
- If the simulation results from existing literature are added, it would involve extra copyright issues.
- If new simulation results are added, there would not be enough context to fully explain the conditions and results of the simulations due to space and contextual limitations.

Despite these limitations, many more explaining descriptions and tables are added in the new manuscript, please refer to the PDF version of this response letter.

Comments 3: Automatic vehicles can work in many spaces, including land, underwater, air, and so on. Are these methods investigated in this article universally applicable? 
Response 3:

It's a great idea to further differentiate the applicable scenarios of the reviewed algorithms. We stressed in Section 2 that the algorithms involved are discussed in the context of autonomous vehicle driving applications.

```
Additionally, these types of algorithms are also used in the trajectory planning of other controlled objects such as unmanned ground vehicles, unmanned surface vehicles, and unmanned aerial vehicles. Due to space limitations, the properties of these algorithms are analyzed and compared only within the scope of autonomous vehicle applications in this article. Furthermore, different types of trajectory planning algorithms are often combined in practical applications to address complex driving scenarios.
```

Reviewer 2 Report

Comments and Suggestions for Authors

The article contributes to the field by presenting an overview of reward learning in automated driving systems, integrating human feedback with diverse sources to enhance trajectory planning. It systematically evaluates different feedback mechanisms and proposes an optimized method for integrating demonstrations and preferences. The research offers significant improvements in the accuracy and efficiency of reward function learning for autonomous vehicles, providing a foundation for future advancements in safe and effective trajectory planning.

It provides a comprehensive overview of various algorithms and methods used in autonomous vehicle trajectory planning and collision risk estimation.

Points to be enhanced or to be corrected

1)The abstract is comprehensive but can be made more concise by focusing on the key contributions and findings. Some sentences are overly complex and could be simplified for better readability.

2)The introduction provides a good overview, but it could benefit from a clearer statement of the paper's motivation. Why is this survey needed now? Highlight the gaps in existing literature more explicitly.

3)The transitions between sections (e.g., from trajectory planning algorithms to collision risk estimation methods) could be smoother. Consider adding brief introductory paragraphs or sentences that connect the topics better.

4)Ensure that section titles accurately reflect the content. For instance, "Trajectory Planning Algorithms in Autonomous Vehicles" could be broken down into more specific subsections (e.g., "Graph Search-Based Algorithms," "Random-Sampling-Based Algorithms") to enhance readability.

5)While the literature review is extensive, ensure that all relevant recent works are included. Some seminal papers might be missing, particularly in the rapidly evolving field of autonomous vehicles.

6)The survey does a good job of comparing different algorithms. However, more detailed comparative tables and figures would help summarize the pros and cons of each approach, making it easier for readers to grasp the key differences quickly.

7)Some algorithms could be explained more intuitively. Including more diagrams, flowcharts, or pseudocode could help readers understand complex concepts.

8)Ensure that all mathematical notations and symbols are consistently defined and used throughout the paper. Some equations and symbols are introduced without adequate explanation.

9)The discussion on integrating collision risk estimation methods with trajectory planning algorithms could be more detailed. How do these methods interact in practical scenarios? Case studies or real-world examples could enhance this section.

10)Consider including more discussion on emerging methods and their potential impact on collision risk estimation. For instance, machine learning-based approaches could be given more attention.

11)The section on user expectations could delve deeper into human factors and psychological aspects. How do different user demographics perceive autonomous vehicle behaviours? Include studies or surveys that address these questions.

12)Discuss how user expectations are currently implemented in commercial autonomous vehicles. Are there any success stories or notable failures?

13) The section "Challenges and Future Prospects" should provide a more detailed analysis of the current challenges faced by the industry. What are the technical, regulatory, and societal hurdles?

14)The conclusion should summarize the key contributions of the paper. Reiterate the main findings and their implications for the field.

Comments on the Quality of English Language

Moderate editing of the English language is required.

Author Response

[Please refer to the PDF version of response letter for the complete content]  

Comments 1: The abstract is comprehensive but can be made more concise by focusing on the key contributions and findings. Some sentences are overly complex and could be simplified for better readability.

Response 1:   Thank you for your advice. We have found that the expressions in the abstract can indeed be further simplified and refined. We have also polished the sentences in other sections of the paper. Below is the revised version of the abstract.

```

Autonomous vehicles are rapidly advancing and have the potential to revolutionize transportation in the future. This paper primarily focuses on vehicle motion trajectory planning algorithms, examining methods for estimating collision risks based on sensed environmental information and approaches for achieving user-aligned trajectory planning results. It investigates different categories of planning algorithms within the scope of local trajectory planning applications for autonomous driving, discussing and differentiating their properties in detail through a review of recent studies. Risk estimation methods are classified and introduced based on their descriptions of sensed collision risks in traffic environments and their integration with trajectory planning algorithms. Additionally, various user experience-oriented methods, which utilize human data to enhance trajectory planning performance and generate human-like trajectories, are explored. The paper provides comparative analyses of these algorithms and methods from different perspectives, revealing the interconnections between these topics. The current challenges and future prospects of trajectory planning tasks in autonomous vehicles are also discussed.

```  

Comments 2: The introduction provides a good overview, but it could benefit from a clearer statement of the paper's motivation. Why is this survey needed now? Highlight the gaps in existing literature more explicitly.

Response 2:   We appreciate your valuable suggestion. We further stressed the gaps in existing literature more clearly in Section 1 in the new manuscript as below.

```

However, the aforementioned reviews exhibit several limitations: they often discuss details in different research cases or types of planning algorithms without thoroughly comparing their characteristics; they address potential risks and issues faced by current autonomous driving systems without detailing how autonomous driving algorithms specifically mitigate these risks; and they emphasize user experience and personalized behavior in autonomous driving systems, yet provide limited discussion on the relationship between driver behavior and planning algorithms. To address these gaps, this paper re-examines the issues of collision risk estimations and user expectations of autonomous driving system behavior from the perspective of trajectory planning algorithm design, and explores the interconnections between these algorithms and methods.

```  

Comments 3: The transitions between sections (e.g., from trajectory planning algorithms to collision risk estimation methods) could be smoother. Consider adding brief introductory paragraphs or sentences that connect the topics better.

Response 3:   We have added some transition sentences in the first version of our manuscript. We further added more transition sentences between paragraphs according to your advice. We believe this point is also correlated with Comments 2, Comments 9, Comments 10, Comments 11 and Comments 12. Therefore, replies and modifications in these points are also considered parts of response to this point.  

Comments 4: Ensure that section titles accurately reflect the content. For instance, "Trajectory Planning Algorithms in Autonomous Vehicles" could be broken down into more specific subsections (e.g., "Graph Search-Based Algorithms," "Random-Sampling-Based Algorithms") to enhance readability.

Response 4:   The titles of the sections and subsections are further adjusted for readability improvements based on your opinion, as shown below.

```  

1. Introduction  2. Trajectory Planning Algorithms in Autonomous Vehicles 2.1. Graph-search-based Trajectory Planning 2.2. Random-sampling-based Trajectory Planning 2.3. Interpolating Curves and Finite-sampling-based Trajectory Planning 2.4. Shooting Numeric Optimization Trajectory Planning 2.5. Collocation Numeric Optimization Trajectory Planning 2.6. End-to-end Data-driven Trajectory Planning 2.7. Spatial and Temporal Spaces, Initial Conditions, and Constraints in Trajectory Planning Algorithms  3. Traffic Environment Collision Risk Estimations for Trajectory Planning Tasks 3.1. Surrogate Safety Measures under Specific Conditions 3.2. Safety Checkers Working in Parallel with Trajectory Planning Algorithms 3.3. Bounding Boxes, Occupancy grids, Reachable Sets, and Driving Corridors 3.4. Potential Fields, Virtual Force Fields, and Composite Risk Fields  4. Human Driver Behaviors and Expected Autonomous Vehicle Trajectory Planning Behaviors 4.1. Mimicking Human Trajectories Directly: Imitation Learning and Data-driven Model Fitting 4.2. Learning from Human Feedback: Preference Learning 4.3. Utilizing Human Behavior Mechanism: Driver Modeling  5. Challenges and Future Perspectives  6. Conclusion

```  

Comments 5: While the literature review is extensive, ensure that all relevant recent works are included. Some seminal papers might be missing, particularly in the rapidly evolving field of autonomous vehicles.

Response 5:   In this review, we have included many up-to-date research references in 2023 and 2024 as listed in the citations. On this basis, we have further added several other studies. We also welcome any other recommendations of relevant papers that can be included in this review article. The figure below demonstrates how many papers are included in this review article by year.  

Comments 6: The survey does a good job of comparing different algorithms. However, more detailed comparative tables and figures would help summarize the pros and cons of each approach, making it easier for readers to grasp the key differences quickly.

Response 6:   Thanks for your advice. We have included a new table to provide a more comprehensive comparison of the different types of trajectory planning algorithms from a macroscopic perspective.

```

Based on all the discussions in Section 2, the properties of these autonomous vehicle trajectory planning algorithm categories can be concluded and compared briefly in Table 6 (in the manuscript). The performance of trajectory planning algorithms is closely related to the application scenario, the specific implementation form, and the details of the algorithm. The following conclusions are qualitative analyses and should not be considered definitive.

```  

Comments 7: Some algorithms could be explained more intuitively. Including more diagrams, flowcharts, or pseudocode could help readers understand complex concepts.

Response 7:   Thanks for your concerns about algorithm details. Below are our concerns regarding this opinion:

- Planning algorithms usually have complex processes, making it difficult to explain them clearly  with a single piece of pseudocode and a small amount of text.

- Different types of planning algorithms are flexible in their application and can be combined with other planning algorithms and autonomous driving algorithm modules in various ways, making it difficult to summarize them with a single flowchart.

- For the above reasons, existing reviews of trajectory planning algorithms usually do not provide specific pseudocode or flowcharts for the algorithms. On the other hand, the flowcharts provided in related review articles are typically for the overall macro-architecture of autonomous driving algorithms. Since this has been extensively discussed, it is not repeated in this paper [citations in the PDF version].  

Comments 8: Ensure that all mathematical notations and symbols are consistently defined and used throughout the paper. Some equations and symbols are introduced without adequate explanation.

Response 8:   Thanks for your concerns about notations and symbols. However, the previous manuscript version does not include specific mathematical notations and symbols because it's a review article. We further make the definitions and abbreviations more clearly demonstrated in the new manuscript.  

Comments 9: The discussion on integrating collision risk estimation methods with trajectory planning algorithms could be more detailed. How do these methods interact in practical scenarios? Case studies or real-world examples could enhance this section.

Response 9:   Thanks for your advice, we added figure of actual case example from existing literature and provided more detailed explanations about how these methods interact in practical scenarios.

```

As autonomous vehicles navigate, they encounter plenty of obstacles within their environment, ranging from various traffic participants and stationary objects to the geometry of the road itself. These obstacles form the navigable areas for autonomous vehicles and pose potential collision risks during transit. Equipped with a diversity of sensors, such as cameras, LiDAR, and millimeter-wave radar, autonomous vehicles can perceive different types of obstacles. Autonomous driving trajectory planning algorithms are imperative to calculate a passable path that circumvents potential collision hazards based on current perceptual data. However, raw sensory information often contains redundancies and lacks direct representation of actual potential risk states. Depending on the characteristics of different types of trajectory planning algorithms, it is essential to transform, fuse, and process these raw sensor data into formats directly applicable to trajectory planning tasks. A concrete example is illustrated in Figure 7 (in the manuscript), where, in a parking scenario, the original point cloud data from LiDAR has been converted into a mathematical representation of reachable sets describing the vehicle's currently accessible area, thereby constraining the state space for the trajectory planning algorithms.

```  

Comments 10: Consider including more discussion on emerging methods and their potential impact on collision risk estimation. For instance, machine learning-based approaches could be given more attention.

Response 10:   Thanks for pointing out the content not covered yet. We added citations and discussions of relevant papers about machine learning-based or data-driven risk estimation approaches. We further found that parts of them are based on motion predictions of traffic participants and usually server as relative independent modules. Therefore, the new discussions are organized as below in Section 3.2.

```

Apart from the rule-based parallel safety checkers discussed above, there are other safety checkers based on the maneuver-oriented motion prediction motion of other traffic participants. Most of these are data-driven approaches utilizing machine-learning algorithms. For example, structured Bayesian networks trained on datasets can infer collision probabilities based on the predicted future states of traffic participants, with these predicted future states obtained from Kalman filter models. Future collision probabilities can also be quantified by heuristic Monte Carlo sampling, with the future states estimated from extended Kalman filtering models. Furthermore, emerging diverse data-driven end-to-end traffic motion prediction approaches provide more reliable future state predictions over longer time ranges compared to traditional Kalman filter models. The prediction results for concerned traffic participants in these approaches are in the forms of behavioral intention classifications, unimodal trajectories, or multimodal trajectories. Specifically, safety checkers based on long-short-term memory networks and graph neural networks provide promising risk estimation results based on predicted trajectories [citations in the PDF version].

```

At the same time, there are other approaches substituting traditional bounding boxes that can be integrated with trajectory planning algorithms, we put the related discussions in Section 3.3.

```

With the development of advanced end-to-end autonomous vehicle perception and prediction algorithms, three-dimensional occupancy grids become promising substitutes for traditional bounding boxes. They provide detailed spacial occupancy information of the obstacles in the environment beyond simple bounding size and position information, providing finer geometry details and possess privileges in describing out-of-vocabulary objects. Currently, the most popular and efficient realizations are based on emerging transformer networks [citations in the PDF version].

```  

Comments 11: The section on user expectations could delve deeper into human factors and psychological aspects. How do different user demographics perceive autonomous vehicle behaviours? Include studies or surveys that address these questions.

Response 11:   That's really a great idea. New discussions and relevant citations are included to  introduce the relationship between autonomous vehicle behaviors and user expectations / acceptance as below.

```

Various survey-based theoretical models, including the Technology Acceptance Model (TAM), Unified Theory of Acceptance and Use of Technology (UTAUT), Theory of Planned Behavior (TPB), and Innovation Diffusion Theory (IDT), have been employed in previous studies to identify the factors influencing user expectations of autonomous vehicles. For instance, studies utilizing TAM have shown that a driver's trust is crucial in shaping their perception of risks, usefulness, and intention to adopt autonomous vehicles. To enhance driver trust, TAM highlights several key aspects: system transparency requires vehicles to demonstrate consistent and predictable behaviors; technical competence demands high performance with minimal errors across various scenarios; and situation management expects the provision of adequate and responsive alternative solutions under the driver’s control. Achieving these aspects necessitates appropriate actions that align with driver expectations, which in turn require a thorough understanding of the complex interactions between traffic participants and the environment. Accordingly, other studies have stated that personalized or human-like driving behaviors in autonomous vehicles have the potential to enhance road safety, transportation efficiency, and human-centric mobility [citations in the PDF version].

```  

Comments 12: Discuss how user expectations are currently implemented in commercial autonomous vehicles. Are there any success stories or notable failures?

Response 12:   Thanks for your advice, we provided new discussions about these topics based on current available information online.

```

Companies like Bosch have already introduced commercial solutions for personalized autonomous driving behaviors, such as Dynamic Distance Assist (DDA). However, the existing solutions can only achieve personalized driving behaviors on relatively basic and well-defined dimensions, such as following distance. The realization of more complex and human-like personalized autonomous driving behaviors remains to be further explored. This requires an in-depth discussion on the utilization of driver behavior data [citations in the PDF version].

```  

Comments 13: The section "Challenges and Future Prospects" should provide a more detailed analysis of the current challenges faced by the industry. What are the technical, regulatory, and societal hurdles?

Response 13:   Thanks for your suggestion, we enriched the contents in challenges and future prospects. The challenge descriptions focus more on technical, regulatory, and societal hurdles.

```

Based on the discussions in the previous sections, the challenges faced by the industry and autonomous vehicles considering trajectory planning tasks can be summarized as follows:

- Technique requirements: The continuously evolving market for autonomous passenger vehicles and robotaxis is placing increasingly higher demands on the performance of autonomous driving techniques. Each type of trajectory planning algorithm has its limitations. Even relatively general trajectory planning algorithms may encounter specific issues in certain driving conditions. Ensuring the robustness and high performance of planning algorithms in complex scenarios with uncertainties from various sources poses a significant challenge. However, these efforts are essential for transportation safety and the widespread adoption of autonomous technology.

- Safety and regulations: The ongoing popularization of autonomous driving technology faces various legal and regulatory restrictions and encounters complex traffic scenarios where autonomous vehicles, traditional manually driven vehicles, pedestrians, and non-motorized vehicles coexist. Autonomous vehicles need to handle complex safety objectives in real-time. Current trajectory planning algorithms primarily utilize fast and direct methods such as bounding boxes, reachable sets, and corridors. Additionally, incorporating considerations for potential risks arising from environmental uncertainties, driver behaviors, and vehicle dynamics into the algorithms is crucial for ensuring road safety.

- User experience and market expectations: The user experience of autonomous driving systems is becoming a key factor influencing their market competitiveness and adoption levels. The current manually set objectives and parameters in trajectory planning algorithms may not align with diverse users' subjective expectations. Methods such as imitation learning, preference learning, and driver behavior modeling have different limitations in addressing this issue. More systematic investigations are needed to develop approaches for designing trajectory planning algorithms oriented towards user experience.

```

The contents in future perspectives are also enriched.

```

Accordingly, considering the challenges mentioned above, potential future perspectives could be outlined below:

- Enhanced Objective Integration: Faced with requirements from planning task constraints and user demands, it is essential to incorporate more comprehensive and detailed objectives concerning various real-world driving tasks into the designs of trajectory planning algorithms. Precisely quantifying these needs through objective metrics and high-quality datasets will make significant differences.

- Advanced Trajectory Planning: As the application scenarios of autonomous driving systems continue to expand and the requirements for takeover rates increase, more advanced trajectory planning algorithms capable of handling complex tasks will emerge. For example, adopting spatio-temporal trajectory planning instead of separate path and velocity planning can yield more optimal results in complex scenarios. Additionally, enhancing consideration of behavioral uncertainty among traffic participants based on available perception and prediction information is a promising development direction.

- Safety and Risk Estimation: Safety and regulatory concerns call for integrating more advanced, realistic yet efficient methods for estimating potential collision risks into planning algorithms to promote safer driving behaviors. For instance, advanced occupancy prediction technologies that provide comprehensive information about the geometries, motions, statuses, and behavioral intentions of traffic participants or other obstacles deserve further investigation. These technologies can be integrated with both end-to-end planning algorithms and traditional modular approaches.

- Human-like and Interpretable Planning: Techniques that generate more consistent, interpretable, and human-like trajectory planning results will attract more attention. It is critical to cultivate human-like driving behaviors that meet user expectations through systematic development of trajectory planning algorithms and diverse parameter tuning methods. Simultaneously, developing more reliable end-to-end trajectory planning methods based on data-driven approaches, while providing more interactions and feedback information, may alleviate user concerns.

```  

Comments 14: The conclusion should summarize the key contributions of the paper. Reiterate the main findings and their implications for the field.

Response 14:   Thanks once again for your detailed review comments, we improved the conclusion part of our paper. The expressions are organized, and the contributions are stressed.

```

This review paper encompasses numerous studies concerning the behaviors of autonomous vehicles and the realization of trajectory planning tasks, covering three main topics: trajectory planning algorithms, collision risk estimation methods involved in these algorithms, and approaches for achieving trajectory planning results that align with user expectations. In summary, the discussed planning algorithms are categorized into various types, including graph-search-based, random-sampling-based, interpolating curves, shooting optimization, collocation optimization, and end-to-end algorithms. Risk estimation methods encompass surrogate safety measures, parallel safety checkers, bounding boxes, occupancy grids, reachable sets, corridors, and various artificial fields. Additionally, user experience-oriented approaches utilizing human data such as imitation learning, preference learning, and driver behavior modeling are explored. This paper provides a comprehensive comparative analysis of algorithms and techniques, uncovering their interconnections between the three topics.

```

Reviewer 3 Report

Comments and Suggestions for Authors

This paper provides a survey of autonomous vehicle behaviors. Trajectory planning algorithms, sensed collision risks and user expectations are introduced. A taxonomy of trajectory planning algorithms for autonomous vehicles are presented, elucidating the distinctive characteristics of various algorithmic types. Collision risk estimation methods are discussed. Diverse approaches are examined to incorporate data derived from human driver behaviors into the trajectory planning algorithm development process. The paper is well written. I only have a few minor comments:

1. In Section 2, graph-search-based, random-sampling-based, interpolating curves, shooting optimization, collocation optimization, and end-to-end algorithms for local trajectory planning tasks of autonomous vehicles are reviewed. It is better to provided a table to show the difference among them.

2. It is better to provide more sentences to show potential future perspectives.

Author Response

[Please refer to the PDF version of response letter for the complete content]

Comments 1: In Section 2, graph-search-based, random-sampling-based, interpolating curves, shooting optimization, collocation optimization, and end-to-end algorithms for local trajectory planning tasks of autonomous vehicles are reviewed. It is better to provided a table to show the difference among them.
Response 1:

Thanks for your advice. We have included a new table to provide a more comprehensive comparison of the different types of trajectory planning algorithms from a macroscopic perspective.
```
Based on all the discussions in Section 2, the properties of these autonomous vehicle trajectory planning algorithm categories can be concluded and compared briefly in Table 6 (in the manuscript). The performance of trajectory planning algorithms is closely related to the application scenario, the specific implementation form, and the details of the algorithm. The following conclusions are qualitative analyses and should not be considered definitive.
```

Comments 2: It is better to provide more sentences to show potential future perspectives.
Response 2:

Thanks for your suggestion. We enriched the future perspective part of our manuscript with more insights based on current technical, regulatory, and societal concerns.
```
Accordingly, considering the challenges mentioned above, potential future perspectives could be outlined below:
- Enhanced Objective Integration: Faced with requirements from planning task constraints and user demands, it is essential to incorporate more comprehensive and detailed objectives concerning various real-world driving tasks into the designs of trajectory planning algorithms. Precisely quantifying these needs through objective metrics and high-quality datasets will make significant differences.
- Advanced Trajectory Planning: As the application scenarios of autonomous driving systems continue to expand and the requirements for takeover rates increase, more advanced trajectory planning algorithms capable of handling complex tasks will emerge. For example, adopting spatio-temporal trajectory planning instead of separate path and velocity planning can yield more optimal results in complex scenarios. Additionally, enhancing consideration of behavioral uncertainty among traffic participants based on available perception and prediction information is a promising development direction.
- Safety and Risk Estimation: Safety and regulatory concerns call for integrating more advanced, realistic yet efficient methods for estimating potential collision risks into planning algorithms to promote safer driving behaviors. For instance, advanced occupancy prediction technologies that provide comprehensive information about the geometries, motions, statuses, and behavioral intentions of traffic participants or other obstacles deserve further investigation. These technologies can be integrated with both end-to-end planning algorithms and traditional modular approaches.
- Human-like and Interpretable Planning: Techniques that generate more consistent, interpretable, and human-like trajectory planning results will attract more attention. It is critical to cultivate human-like driving behaviors that meet user expectations through systematic development of trajectory planning algorithms and diverse parameter tuning methods. Simultaneously, developing more reliable end-to-end trajectory planning methods based on data-driven approaches, while providing more interactions and feedback information, may alleviate user concerns.
```

Round 2

Reviewer 2 Report

Comments and Suggestions for Authors

The reviewer’s comments have been fully responded and the paper has been substantially revised.

I do not have further comments.

Thank you

Comments on the Quality of English Language

 Minor editing of the English language is required.